# Pre-Senescence Induction in Hepatoma Cells Favors Hepatitis C Virus Replication and Can Be Used in Exploring Antiviral Potential of Histone Deacetylase Inhibitors

**DOI:** 10.3390/ijms22094559

**Published:** 2021-04-27

**Authors:** Alsu Z. Malikova, Anastasia S. Shcherbakova, Konstantin A. Konduktorov, Anastasia S. Zemskaya, Alexandra A. Dalina, Vladimir I. Popenko, Olga G. Leonova, Alexei V. Morozov, Nikolay N. Kurochkin, Olga A. Smirnova, Sergey N. Kochetkov, Maxim V. Kozlov

**Affiliations:** Engelhardt Institute of Molecular Biology, Russian Academy of Sciences, 119991 Moscow, Russia; rasssl44@gmail.com (A.Z.M.); nas.sherbakova@yandex.ru (A.S.S.); kkonduktorov@yandex.ru (K.A.K.); a.zemskaia@mail.ru (A.S.Z.); alexandra.dalina@gmail.com (A.A.D.); popenko@eimb.ru (V.I.P.); leonova-kozma@mail.ru (O.G.L.); runkel@inbox.ru (A.V.M.); nikola.76@mail.ru (N.N.K.); o.smirnova.imb@gmail.com (O.A.S.); snk1952@gmail.com (S.N.K.)

**Keywords:** hepatitis C virus, TGF-β1, palbociclib, pre-senescence, DAAs, HTAs, HDACi

## Abstract

Recent evidence suggests that fibrotic liver injury in patients with chronic hepatitis C correlates with cellular senescence in damaged liver tissue. However, it is still unclear how senescence can affect replication of the hepatitis C virus (HCV). In this work, we report that an inhibitor of cyclin-dependent kinases 4/6, palbociclib, not only induced in hepatoma cells a pre-senescent cellular phenotype, including G1 arrest in the cell cycle, but also accelerated viral replicon multiplication. Importantly, suppression of HCV replication by direct acting antivirals (DAAs) was barely affected by pre-senescence induction, and vice versa, the antiviral activities of host-targeting agents (HTAs), such as inhibitors of human histone deacetylases (HDACi), produced a wide range of reactions—from a dramatic reduction to a noticeable increase. It is very likely that under conditions of the G1 arrest in the cell cycle, HDACi exhibit their actual antiviral potency, since their inherent anticancer activity that complicates the interpretation of test results is minimized.

## 1. Introduction

The success achieved in the modern therapy of hepatitis C due to the use of direct acting antivirals (DAAs) provides time to find new approaches to the treatment of this disease. The development of drugs directed against host targets (HTAs) involved in the reproduction of hepatitis C virus (HCV) provides new opportunities to combat this pathogen and overcome future challenges associated with the emergence of resistance to DAAs [1].

Despite the significant number of clinical trials for various HTAs, none of them have been approved for practical use, which is likely due to: (*i*) the complex physiological effects of such compounds [2], and (*ii*) the absence of adequate cell-based test systems [3]. The HCV replicon test system that has proven to be extremely useful for searching for DAAs and for the initial screening and optimization of DAAs does not guarantee that identified HTA hits will be suitable for in vivo applications [4]. To improve the efficiency of HTA screening, the functional status of hepatoma cells in the test system must be obviously brought closer to the status of chronically HCV-infected hepatocytes.

In practice, it is the chronic form of hepatitis C (CHC) that most often requires treatment [5], when two pathological processes occur simultaneously—the viral infection itself and the senescence-associated chronic inflammatory response caused by it [6,7]. The scattered data derived from the analysis of liver tissue and sera of patients with CHC assemble into a puzzle of the overall expansion of cellular senescence as liver injury intensifies. An increase in the concentration of reactive oxygen species (ROS) and induction of DNA damage [8,9], secretion of proinflammatory cytokines and growth factors [10,11,12], G1 arrest in the cell cycle [13], development of insulin resistance [14], and, finally, epigenetic modifications of DNA and histones [15,16] altogether constitute the “classical list” of factors that trigger senescence or comprise the senescence signature [17].

Cellular senescence is activated for the universal purposes of the organism’s antiviral defense [18], but in some cases, cellular replicative senescence can contribute to increased viral replication [19]. In chronically infected hepatocytes, HCV provokes oxidative stress-dependent DNA damage and simultaneously inhibits the expression of DNA repair enzymes [20,21], which gradually and inevitably provides affected cells with a senescent status [22,23,24]. In addition, senescent hepatocytes can accumulate in damaged liver tissues as a result of a process known as “senescence-induced senescence” [25].

Senescent cells communicate with each other and with the local environment, secreting inflammatory cytokines, chemokines, growth factors, and matrix-modifying enzymes, thus contributing to the senescence-associated secretory phenotype (SASP) [26]. Transforming growth factor-β1 (TGF-β1) is part of the SASP and ensures the irreversibility of G1 arrest in the cell cycle via Smad3/4-dependent induction of the expression of CDK inhibitors [27]. Immunohistochemical analysis of biopsies from chronic hepatitis C patients had shown a sharp increase in the concentration of TGF-β1 in periportal hepatocytes and hepatic sinusoidal cells in comparison with control samples from healthy individuals [11,28]. It should be noted that high external concentrations of TGF-β1 have caused senescence in cultures of well-differentiated cells of hepatocellular carcinoma [29].

Palbociclib (PD0332991) is a cell-permeable inhibitor of CDK4/6 that occupies the ATP-binding pocket of the enzyme, thereby preventing retinoblastoma protein (RB) phosphorylation and cell growth by inducing G1 arrest [30,31]. There is a remarkable similarity in the proliferation suppression and senescence induction effects of TGF-β1 [29] and palbociclib [32] on human liver cancer cell lines, especially with respect to the Huh7 cell line commonly used for HCV replicon transfection studies. Both TGF-β1 and palbociclib have blocked RB phosphorylation while simultaneously decreasing p107 and increasing p103 total levels. These changes in the RB family proteins correlate with the exit from the cell cycle [33]. In addition, p21 accumulation has been detected after exposure to either palbociclib or TGF-β1, despite the unchanged total level of p53 [29,32].

In the present work, we explored the feasibility of using palbociclib as a low-molecular weight analogue of TGF-β1 to induce senescence in hepatoma cells and to study how the pre-senescent status of the cells affects HCV replication and the efficacy of representative anti-HCV agents. 

## 2. Results

### 2.1. TGF-β1 and Palbociclib Affect HCV Replication and Cell Viability in Different Manners

For an initial assessment of the effect of TGF-β1 and palbociclib on replicon functioning in the Huh7-luc/neo cells, we tested both agents, as it is conducted when screening compounds for antiviral activity. The incubation of cells in the presence of TGF-β1 or palbociclib led to a reduction in the luciferase signal (which indicates the presence of the replicon) in both cases but in different manners. The replicon concentration curve monotonically decreased for TGF-β1 (Figure 1A), but not for palbociclib, for which an extended plateau of the luciferase signal was observed in the concentration range of 0.3–3 μM at approximately 50% of the initial value detected in the absence of the drug (Figure 1D).

The next step was the measurement of the cytotoxic/cytostatic effects of both substances. In the case of TGF-β1, the dose–response curve for cell viability was inverse S-shaped, reaching a gradually sloping plateau of the MTT signal (produced by living cells) in the concentration range of 0.1–1 ng/mL (Figure 1B). In the case of palbociclib, the curve looked very different. After an initial decrease, the presence of the inhibitor in the concentration range of 0.3–3 μM did not significantly affect cell viability that remained at about 60% of the original value (Figure 1E). To discriminate the cytotoxic/cytostatic effects observed in the MTT assay, the proportion of damaged cells was assessed by the values of propidium iodide uptake [34]. We found that with the increase in the concentration of palbociclib from 0.3 to 10 µM, the proportion of dead cells monotonically increased in comparison with the untreated control from 10.7% to 21.9%, respectively (Appendix A). Thus, the cytostatic effect of the inhibitor in the same concentration range led to an additional drop in the MTT signal by approximately 20–25% (Appendix A).

When screening low-molecular weight compounds for antiviral activity, the level of the luciferase signal in the lysate of the Huh7-luc/neo cells is most often a reliable indicator of the HCV RNA content [35]. However, the complex effects of TGF-β1 and palbociclib on cellular signaling pathways required an additional determination of the viral RNA level in order to obtain more accurate information about the replicon operation. We found that in the presence of both TGF-β1 and palbociclib, the balance between the levels of replicon and luciferase changed significantly. In the case of TGF-β1, the viral RNA level curve was inverse S-shaped, while the normalized luciferase signal curve paralleled the shift towards lower concentrations of the growth factor (Figure 1C).

On the contrary, in the presence of palbociclib in the concentration range of 0.3–3 μM, the viral RNA level increased sharply, while the normalized luciferase signal displayed a pronounced plateau (Figure 1F). Thus, in comparison with the untreated control, both TGF-β1 and palbociclib increased the replicon level in relation to the luciferase signal by 2–2.5 times. This result may partly be explained by the accelerated degradation of luciferase. Indeed, since the reporter firefly luciferase is bound to ubiquitin in the fusion protein translated from the first cistron of the replicon [36], the ubiquitin-dependent proteasomal degradation pathway for luciferase appeared most preferable. Interestingly, according to the published data, palbociclib is able to induce proteasome activation mediated through the reduced proteasomal association of the ECM29 protein [37].

Next, we established how palbociclib affects the stoichiometric ratio between HCV RNA and viral enzymes—NS3 protease/helicase and NS5B RNA-dependent RNA polymerase—that are translated from the second cistron of the replicon. Based on the results of Western blot analysis (Figure 1G), we constructed the corresponding NS3/α-Tub and NS5B/α-Tub dependencies which indicated an increase in the levels of both proteins in the presence of palbociclib in the concentration range of 0.3–3 µM (Figure 1H). However, in order to correctly compare the levels of luciferase and viral proteins NS3 and NS5B, it was necessary to perform additional normalization by calculating the increase or decrease in the protein concentrations relative to the common denominator, the replicon content. Accordingly, in the inhibitor concentration range of 0.3–3 μM, we obtained U-shaped curves of the luciferase level (0.4 min) and NS3 level (0.8 min), and in the case of NS5B, after an initial drop, the curve reached a gradually sloping plateau at the level of 0.75–0.70 (Figure 1I). Then, using affinity labeling of proteasomes with a fluorescent analogue of the substrate followed by cytofluorometry, we found that the incubation of cells with 0.3 µM palbociclib increased the number of active proteasomes by 1.6 times, but a further increase in the concentration of the inhibitor eliminated the initial effect (Figure 1I).

The mirror-symmetric appearance of the proteasome activation curve versus the curves of luciferase and NS3 levels (Figure 1I) indicated that for both of these proteins, the proteasomal degradation pathway was the main one. The half-lives of NS3 and NS5B are known from the literature to be almost the same—14.5 h and 12 h, respectively [38]. To determine the preferred degradation pathways of NS3 and NS5B in the Huh7-luc/neo cells, we alternately halted in them total protein synthesis, proteasomal protease activity, and total lysosomal activity using cycloheximide, MG132, and hydroxychloroquine, respectively (Figure 1J). The 18-h incubation of cells in the presence of cycloheximide was shown to result in a sharp reduction in the levels of both proteins. During this time, NS3 accumulated mainly if proteasome activity was inhibited, while the NS5B content increased equally when either proteasomal or lysosomal protease activity was suppressed.

### 2.2. Palbociclib Effectively Converts Cells to a Pre-Senescent State

Proliferation arrest is the main hallmark of cellular senescence activation. As evidenced by flow cytometry, treatment of cells with 1 μM palbociclib as early as 24 h caused an extraordinary accumulation of cells arrested in the G1 phase in the cell cycle (Figure 2A). In the presence of TGF-β1, a similar redistribution of cells among the phases of the cell cycle was achieved 24 h later and at a TGF-β1 concentration of 10 ng/mL that is incompatible with the survival of the HCV replicon, as previously shown (Figure 1A).

The TGF-β1-triggered senescence is associated with the induction of Nox4 and the accumulation of ROS [29]; in turn, the CDK4/6 inhibition mediated by palbociclib also induces ROS [39]. Judging by the amplification of the *DCF* fluorescent signal, the ROS level in cells treated with either the growth factor or the inhibitor increased approximately on the same scale; moreover, the co-treatment of the treated cells with 10 mM NAC as an ROS inhibitor significantly suppressed the flare-up of fluorescence, which confirmed the specificity of the observed effect (Figure 2B).

As senescence progresses, cells undergo morphological changes—the cell nucleus and overall size of cells increase, and their shape becomes flattened. In addition, the number and size of lysosomes that have an elevated senescence-associated β-galactosidase activity (SA-β-Gal) sharply increase in cells [40]. Staining of the Huh7-luc/neo cells with *Acridine O.*, which provides nuclei with green and lysosomes with red fluorescence, showed that 48 h of incubation of cells with TGF-β1 or palbociclib is sufficient to activate biogenesis of lysosomes of predominantly perinuclear localization (Figure 2C).

After entering the cell, palbociclib concentrates in acidic vesicles [41], where weakly basic residues of the drug, piperazine (pKa of 7.4) and pyridine (pKa of 3.9), become protonated [42], which prevents its return transfer into the cytosol (so-called lysosomal trapping). Simultaneous selective fluorescent staining of cells with *Hoechst* (blue signal in nuclei), *LTR* (red signal in lysosomes), and palbociclib (green signal) became feasible due to the felicitous combination of spectral characteristics of all three compounds. Using this circumstance, we identified the colocalization of *LTR* and palbociclib in the Huh7-luc/neo cells and found that efficient lysosomal trapping occurs in the inhibitor concentration range of 1–10 μM (Figure 2C).

SA-β-Gal activity is one of the most widely used markers for the identification of senescence in cells and tissues [40]. To assess the time frame in which the transition from pre-senescence to senescence occurs in Huh7-luc/neo cells treated with palbociclib, we stained the cells with *X-Gal*, the chromogenic substrate for β-galactosidase. After three days, the size of the cells increased, and they became more rounded and flattened, but the activity of β-galactosidase remained at the control level, indicating a pre-senescent status of the cells (Figure 2D). After seven days of incubation, the treated cells increased in size even more and were intensely stained in the presence of *X-Gal*, confirming their transition to the senescence state. It is worth noting that cell confluence decreased sharply as either the concentration of palbociclib or the duration of incubation with it increased, clearly illustrating the strong cytostatic as well as cytotoxic effects of the drug.

### 2.3. Palbociclib Causes Accumulation of Lipid Droplets and Blocks Lipophagy Flux

Abundance and increased size of lipid storage droplets are characteristic signs of cellular senescence [43]. Staining of the Huh7-luc/neo cells with *LTR* and *Bodipy*, a green fluorescence tracer of LDs, showed that in the presence of TGF-β1 and palbociclib, the area occupied in the cell by both lysosomes and lipid droplets increased; the fluorescence intensity of stained structures increased as well (Figure 3A,D,E). However, the colocalization of the red and green signals in both cases was fundamentally different.

In the presence of 3 ng/mL TGF-β1, some LDs were stained orange (Figure 3B). Compared to the untreated control, the values of Pearson’s colocalization coefficient (PCC) increased by many times from 0.00–0.02 to 0.32 (Figure 3C, T3). The highest colocalization of LDs and lysosomes was observed in those regions of the cytoplasm where large lysosomes were located. In such regions, PCC was 0.702. Apparently, the number of organelles with mixed orange fluorescence (*LTR* + *Bodipy*) was increased as a result of the fusion of lysosomes with autophagosomes sequestering LDs by lipophagy. However, this fusion was not total. In the profile of the distribution of green and red signals in a cell treated with 3 ng/mL TGF-β1 (Figure 3B), in addition to well-coinciding peaks, there are green peaks that do not overlap with the signal from lysosomes.

Although in the presence of palbociclib, the area occupied by LDs and lysosomes in the cytoplasm sharply increased, no signs of fusion between these organelles were observed. The value of Pearson’s correlation coefficient at a palbociclib concentration of 10 μM was at the level of the untreated control and was only 0.018 (Figure 3C, P10). Accordingly, there was no coincidence of peaks on the profile of the distribution of green and red signals (Figure 3B). In general, it can be assumed that the absence of colocalization of LDs and lysosomes in the treated cells was a consequence of deviation from the normal course of lipophagy at the stage of fusion between these organelles.

Using Western blot analysis, we found that the impairment of lipophagy was not caused by inhibition of autophagosome formation [44]; on the contrary, palbociclib facilitated, in a dose-dependent manner, the accumulation of the phosphatidylethanolamine conjugate of the LC3 protein (LC3-II) which is responsible for autophagosome expansion, sealing, and functioning (Figure 4A).

The “expendable” autophagy protein p62/SQSTM1, a receptor that ensures selectivity of “cargo” delivery into autophagosomes, is involved in LD turnover through interaction with the LD membrane protein ADPR [45]. In the palbociclib concentration range of 0.3–10 μM, the levels of both p62 and LC3-II gradually increased (Figure 4A). The synchronous increase in the levels of these proteins most likely indicates an incomplete progress of autophagy, which well corresponds to the inhibition of lipophagy in the presence of palbociclib (Figure 3B). On the contrary, in the presence of TGF-β1 in the concentration range of 0.3–3 ng/mL, the p62 level decreased with a simultaneous increase in the LC3-II content (Figure 4A,B), which is in good agreement with the active course of lipophagy that was also identified by microscopy (Figure 3B).

To confirm activation of autophagosome biogenesis in the presence of TGF-β1 and palbociclib, we generated Huh7 cells stably expressing GFP-LC3 using the lentiviral infection system. The maximum cell transfection efficiency after one round of selection was approximately 60% (Figure 3C). At the baseline autophagy level, the transfected cells displayed a well-defined but diffuse cytoplasmic signal of the GFP-LC3 fusion protein, while in the presence of TGF-β1 and palbociclib, the fluorescent signal became brighter and more discontinuous in a dose-dependent manner, which indicated an increase in expression of GFP-LC3 and its re-localization from the cytosol to autophagosomal membranes (Figure 4E).

### 2.4. Comparative Testing of DAAs and HDACi under Regular and Pre-Senescence Conditions

The fact that the development of pre-senescence in the Huh7-luc/neo cells had a stimulating effect on viral replication became a prerequisite for the creation of a test system for the screening of compounds under conditions characteristic of cellular senescence. The preliminary 24-h incubation in the presence of 1 μM palbociclib increased the concentration of ROS in the cells and converted them into the G1 arrest state. Testing of the compounds was conducted in the next 48 h, also in the presence of 1 μM palbociclib. During this period, the morphology of the cells changed, and the formation of lysosomes, autophagosomes, and lipid droplets was actively taking place in them, although the progress of autophagy was incomplete and lipophagy was blocked.

To establish how the pre-senescence state of the Huh7-luc/neo cells could affect the antiviral action of DAAs, we chose four compounds: telaprevir [46], an inhibitor of viral protease NS3, and nucleoside/nucleotide analogues 2′-Me-Ad [47], sofosbuvir [48], and ribavirin [49] (Table 1). The dose response of replicon inhibition by telaprevir and 2′-Me-Ad was almost the same in both regular and reprogrammed test systems, as would be expected for DAAs, the inhibition efficiency of which against viral enzymes is independent of the cell status (Table 1). Surprisingly, in pre-senescent cells, the anti-HCV activity of sofosbuvir and ribavirin moderately decreased by 2.5 and 3 times, respectively. Perhaps, in the case of sofosbuvir, this was due to a slowdown in the multi-stage activation of the metabolic prodrug [48], or, in the case of ribavirin, due to a pleiotropic rather than a direct antiviral mechanism of action [49].

HDACi are a promising class of anti-HCV agents, some of which inhibit the multiplication of HCV in vitro and in vivo [50,51], while others slow down the development of hepatocellular carcinoma [52]. Potentially, such a combination of activities in one compound may provide an advantage in the treatment of the late stages of CHC accompanied by carcinogenesis [53], but, on the other hand, it also complicates HDACi testing in regularly proliferating hepatoma cells.

Next, we applied the two-stage testing procedure (with or without palbociclib) for histone deacetylase inhibitors (HDACi), the selective action of which covered the entire spectrum of zinc-dependent histone deacetylase classes I, IIa, and IIb. In total, we tested: (*i*) multipotent inhibitors—vorinostat and belinostat [53,54]; (*ii*) HDAC1/2/3 and HDAC8 class I inhibitors—CI-994, SR-3212, and *ortho-*PhO-CHA [50,55,56]; (*iii*) HDAC4/5/7/9 class IIa inhibitors—Cmpd13 and TMP-269 [54]; (*iv*) an HDAC6 class IIb selective inhibitor—Cmpd12a; and (*v*) HDAC6/10 class IIb dual inhibitors—tubastatin A and bufexamac [54,57,58] (Table 2).

The use of pre-senescent cells for testing HDACi resulted in a dramatic decrease in the anti-HCV activities of vorinostat, belinostat, and Cmpd12a, convincingly demonstrating that the senescence status of the cells can significantly affect the antiviral potency of HDACi (Table 2). The dose response of replicon inhibition by CI-994 and bufexamac or *ortho-*PhO-CHA, TMP-269, and tubastatin A was the same or even more sensitive in the reprogrammed test system compared to the regular one, indicating that HDAC classes I/IIa/IIb (except HDAC6) remain vulnerable anti-HCV targets under pre-senescence conditions. It should be noted that the anti-HCV activities of SR-3212 and Cmpd13 decreased, although moderately (by 2 and 3 times, respectively).

### 2.5. Significance of HDAC6 as Anti-HCV Target Is Reduced in Pre-Senescent Hepatoma Cells

The significant discrepancy between the replicon inhibition in pre-senescent cells by Cmpd12a and tubastatin A clearly indicates that the mechanisms of the antiviral action of these compounds differ substantially. One of the possible reasons for the decrease in the efficiency of replicon inhibition by Cmpd12a under pre-senescence conditions may be the HDAC6 overexpression induced by Cmpd12a but not by tubastatin A. To explore this possibility, the relative level of HDAC6 activity in the presence of both compounds in regular and palbociclib-treated cell cultures was determined at various concentrations of inhibitors. α-TubulinK40ac is the main cytoplasmic substrate for HDAC6, the inhibition of which results in an increase in tubulin acetylation in vivo [59]. Western blot analysis of the relative abundance of Ac-α-tubulin/α-tubulin in regularly proliferating (RP) or in pre-senescent (PS) cells showed that the degree of inhibition of the HDAC6 deacetylation activity by tubastatin A as well as by Cmpd12a was mostly equal in each case (Figure 5A).

It has been reported that depletion or inhibition of HDAC10 results in the accumulation of lysosomes in neuroblastoma cell lines [57]. In line with these data, staining the Huh7-luc/neo cells with *Acridine O.* showed that tubastatin A, a dual inhibitor of HDAC6/10, induces lysosomal accumulation, while Cmpd12a, a selective inhibitor of HDAC6, does not. We also found that Cmpd12a does not prevent an increase in the area occupied by lysosomes in the presence of palbociclib (Figure 5B).

## 3. Discussion

HCV infection provokes in affected hepatocytes of liver tissue the development of cellular senescence features that paradoxically accompany the multiplication of the virus at the later stages of the disease, while at the early stages, they correlate with the activation of immune defense mechanisms of the organism. Most likely, the progression of senescence is a consequence of chronic inflammation caused by HCV infection, in which senescent hepatocytes can accumulate both as a result of the direct action of the virus and due to paracrine transmission of the senescence signal by SASP components. The problem of the reproduction of the virus in senescent hepatocytes has not yet been studied, although it may be of fundamental importance for understanding the dynamics of the development of infection and its treatment at the most severe stages of the disease [60].

The commonly used HCV replicon test systems for antiviral compound screening take advantage of rapidly proliferating cells of hepatocellular carcinoma, which leads to detection of false positives that block cell division rather than inhibit viral replication. The approach proposed in this study is based on pre-senescence induction in replicon-harboring cells; this may help to resolve the issue by making it possible to reliably distinguish between antiviral and anticancer effects of HTAs represented in this work by histone deacetylase inhibitors.

Numerous data indicate that viral RNA replication and viral particle assembly occur in organelle-like HCV replisomes which are lipid droplets decorated by double-membrane vesicles (DMVs) [61]. DMVs are formed at the early stages of autophagy (triggered by HCV infection) and are likely to not contain the LC3 protein required for autophagosome maturation and functioning [62]. The close proximity of sites implicated in replication and assembly guarantees maximum safety of viral RNA, while lipid droplets provide the virus with the machinery necessary for assembly of lipid virus particles [63]. Note that the activation of biogenesis of LDs is also a characteristic feature of cellular senescence [43].

Despite the attractive possibility to use externally added TGF-β1 to induce senescence in hepatoma cells, our data show that TGF-β1 concentrations that are not harmful to the replicon do not cause the G1 arrest in the cell cycle. Additionally, though there were signs of pre-senescence induction, including the formation of ROS and accumulation of lysosomes and lipid droplets, it is likely that the activation of lipophagy led to a sharp reduction in the replicon content.

Fortunately for us, the use of palbociclib, a CDK4/6 inhibitor, resolved the issue. This drug caused an accelerated development of pre-senescence features in the Huh7-luc/neo cells, including the G1 arrest in the cell cycle, and led to an increase in the content of viral RNA and proteins. This may be the result of an active formation of replisomes, accompanied by prevention of their lysosomal digestion. This assumption is supported by our finding of the accelerated autophagosome biogenesis and accumulation of LDs, accompanied by the complete absence of fusion of lipid-containing organelles with lysosomes. It is worth noting that the action of palbociclib in this aspect is very similar to autophagy inhibition by chloroquine [64]. This is probably a consequence of the similarity of the molecular structure of both compounds, where the heterocyclic nucleus and the alkylammonium residue are linked together through a short linker. Overall, the use of palbociclib allowed us to mimic the effect of TGF-β1 in respect to senescence induction and to avoid major drawbacks of TGF-β1, thus providing the first functional model for HCV-infected pre-senescent hepatocytes.

Comparative testing of antiviral activities of compounds in RP and PS hepatoma cells was originally conducted for two DAAs—telaprevir, a peptidomimetic inhibitor of the HCV NS3-4A serine protease [46], and 2′-Me-Ad, a precursor of 2′-Me-ATP, that targets viral transcription, causing premature termination of the growing RNA chain [47]. Using these reference compounds, we were able to establish a guideline for comparing results obtained with regular and reprogrammed test systems. Indeed, since the inhibition efficiency of telaprevir or 2′-Me-ATP against the viral NS3-4A serine protease or the NS5B RNA polymerase is independent of the cycle stage, and since pre-senescence has some positive effect on HCV replication (as discussed above), the EC_50_ values obtained in both test systems should be comparable. In accordance with this assumption, the antiviral potencies of telaprevir and 2′-Me-Ad determined in both assays were nearly identical.

The significant reduction in anti-HCV activity of the next two tested DAAs, sofosbuvir and ribavirin, in pre-senescent cells was probably associated with the influence of cellular factors. Indeed, in the case of sofosbuvir, the conversion of a phosphoramidate prodrug into an intermediate nucleoside-5′-monophosphate is carried out by the cellular enzymes cathepsin A (CatA) and histidine triad nucleotide-binding protein 1 (Hint1) [65]. Thus, the antiviral effect of sofosbuvir depends on the activities of these enzymes and is not direct in the sense of the action of telaprevir or even 2′-Me-Ad. A similar conclusion can be made for ribavirin, the antiviral effect of which, in addition to the direct inhibition of HCV replication, involves the inhibition of the host inosine monophosphate dehydrogenase (IMPDH) enzyme [49].

The validity of the developed comparative assay for the antiviral activity of DAAs was confirmed on HTAs—inhibitors of HDACs. Our previous results clearly demonstrated that different HDACi targeting HDAC classes I, IIa, and IIb effectively suppress HCV replication in regularly proliferating hepatoma cells [54,56]. Determination of the antiviral activity of a representative set of HDACi in the reprogrammed test system showed that HDAC classes I/IIa/IIb retain their role in viral replication, with the exception of HDAC6, the value of which as a therapeutic target appears to be significantly lower.

Regularly proliferating hepatoma cells harboring the HCV replicon are obviously a too complex system for the interpretation of the antiviral effect of vorinostat and belinostat because both compounds possess an anticancer activity affecting fast-growing cells. The use of pre-senescent cells in the test system eliminated this problem due to the absence of proliferation, which ultimately made it possible for us to adequately measure the anti-HCV activity of these compounds and assess their antiviral potency under pre-senescence conditions, which was unfortunately found to be significantly weakened.

Earlier, using tubastatin A as an HDAC6 inhibitor, we showed the concurrence of the curves of replicon inhibition and accumulation of the acetylated form of α-tubulin (α-TubK40ac), for which HDAC6 is the main deacetylating enzyme in Huh7-luc/neo cells [66]. In this work, using WB analysis of the α-TubK40ac content in RP and PS cells after their treatment with tubastatin A as well as Cmpd12a, we found similar dependences of the α-TubK40ac accumulation on the inhibitor concentration in both cases, but in contrast to tubastatin A, the anti-HCV activity of Cmpd12a was significantly weakened under the conditions of pre-senescence.

The discovered difference clearly indicates that the mechanisms of the antiviral action of these compounds substantially differ. It has recently been shown that the tubastatin A scaffold ensures simultaneous binding to HDAC6 and HDAC10 [67]. The affinity of tubastatin A binding to the active site of HDAC10 is ensured by the interaction of the tertiary amino group of the inhibitor’s *cap* with the carboxyl group of the Glu272 residue of the enzyme. The presence of tubastatin A in the Huh7-luc/neo cells led to accumulation of lysosomes as a visible result of the inhibition of HDAC10, which has previously been described for neuroblastoma cells [57]. As a neutral structural analogue of tubastatin A, Cmpd12a is not capable of forming the corresponding ion pair and therefore does not bind to HDAC10. The participation of HDAC10 in the HCV life cycle was previously unknown; however, an important consequence of the simultaneous inhibition of HDAC6 and HDAC10 by tubastatin A is the preservation of the antiviral activity of the drug during the conversion from RP to PS hepatoma cells. Thus, we were able, for the first time, to suggest HDAC10 as a vulnerable target for anti-HCV treatment, and although these results are quite preliminary, they demonstrate the unique possibilities of the proposed approach for studying and assessing the antiviral potential of HDACi.

## 4. Materials and Methods

### 4.1. Cell Cultures and Plasmids

In line with an earlier publication [68], the Huh7 cell line was grown in the medium DMEM + F12 (2:1) supplemented with 10% FBS (HyClone, Logan, UT, USA), 2 mM l-glutamine, 100 U/mL penicillin, and 100 μg/mL streptomycin in the presence of 5% CO_2_ at 37 °C. Cells were re-seeded every three days at the ratio of 1:3 or 1:5. The Huh7-luc/neo culture was grown at the same conditions with the addition of 330 μg/mL geneticin (G418 sulfate, Selleckchem, Houston, TX, USA). Huh7-luc/neo cells harboring the luciferase reporter genomic HCV replicon genotype 1b were a generous gift from Prof. Ralf F.W. Bartenschlager (University of Heidelberg, Heidelberg, Germany). The reporter plasmid pLCMV-GFP-LC3B-puro and packaging plasmids for lentiviral production were kindly provided by Prof. Peter M. Chumakov (Engelhardt Institute of Molecular Biology of RAS, Moscow, Russia).

### 4.2. Reagents and Antibodies

Chemicals: palbociclib hydrochloride (S1116), telaprevir (S1538), sofosbuvir (S2794), ribavirin (S2504), vorinostat (S1047), belinostat (S1085), CI-994 (S2818), TMP-269 (S7324), tubastatin A (S8049), and bufexamac (S3023), were all from Selleckchem (Houston, TX, USA); 2′-Me-Ad, SR-3212, Cmpd13, *ortho*-PhO-CHA, and Cmpd12a were synthesized as previously described [54,55,56,69,70]. The stock solutions of the tested compounds (usually 100 mM in DMSO) were serially diluted with ethanol to 100× concentrations; the corresponding 100× solution was added to each well of a culture plate in an amount of 1% of its volume. *Vehicle control* samples were treated with the same amount of ethanol. Dyes: X-Gal (R0404) was from Thermo Scientific (Waltham, MA, USA); Hoechst 33342 trihydrochloride trihydrate (H3570), Acridine Orange (A1301), LysoTracker Red DND-99 (L7528), and Bodipy 493/503 (D3922) were all from Invitrogen (Waltham, MA, USA). ROS reagent 2′,7′-dichlorofluorescein diacetate **(**D6883) was from Sigma (St. Louis, MO, USA). Fluorescent proteasome probe Me4BodipyFL-Ahx3Leu3VS (UbiQ-018) was from UbiQ, the Netherlands.

TGF-β1 protein (ab116150) was from Abcam (Cambridge, MA, USA). An amount of 5 μg of lyophilized product was reconstituted in 20 μL of 10 mM; sodium citrate (pH 3.0), kept for 5 min, diluted with 80 L of 0.2% BSA in PBS, divided into 10 μL aliquots, and stored at −20 °C.

The following antibodies were used: mouse anti-NS3 (LS-C343002) was from LSBio (Seattle, WA, USA); rabbit anti-NS5B (ab35586), rabbit anti-LC3B (ab51520), mouse anti-SQSTM1/p62 (ab56416), and rabbit anti-acetylated lysine (ab80178) were from Abcam, USA; mouse anti-α-tubulin (T5168) was from Sigma, USA. Conjugates of horseradish peroxidase with secondary specific antibodies (anti-mouse (sc-2005) and anti-rabbit (sc-2004)) were from SCBT (Dallas, TX, USA).

### 4.3. HCV Replicon Assay

In line with an earlier publication [68], Huh7-luc/neo cells were seeded at a density of 1 × 10^4^ cells/well into 48-well culture plates (without antibiotic G418). After 24 h (30–40% monolayer), different concentrations of the test substances were added to the culture medium. After the incubations, the medium was removed, and cells were washed with PBS and lysed. Luciferase activity of the reporter protein was measured using a Luciferase Assay System Kit (Promega, Madison, WI, USA) according to the manufacturer’s protocol.

### 4.4. Cell Viability Assay

Huh7-luc/neo cells were seeded at a density of 3.5 × 10^3^ cells/well into 96-well culture plates (without antibiotic G418); 24 h after seeding (30–40% monolayer), cells were treated with various concentrations of the test substances for an appropriate time. At the end of the treatment, cell viability was determined using the Cell Proliferation Kit I (MTT assay) as specified by the manufacturer (Sigma-Aldrich, St. Louis, MO, USA).

### 4.5. Measuring Cell Death 

Huh7-luc/neo cells were seeded at a density of 1 × 10^5^ cells/well into 6-well culture plates (without antibiotic G418), cultured for 24 h (30–40% monolayer), and then treated with different concentrations of palbociclib for 48 h. The determination of cell death by propidium iodide uptake and flow cytometry was performed as previously described [34].

### 4.6. Pre-Senescence Induction

Huh7-luc/neo cells were seeded as described above, cultured for 24 h, and incubated with 1 µM palbociclib for an additional 24 h. Then, cells were treated with different concentrations of the test substances in the presence of palbociclib for the specified time. Methods for measuring and calculating EC_50_ and CC_50_ values for DAAs and HDACi in pre-senescent cells were exactly the same as in the case of regularly proliferating cells.

### 4.7. Real-Time PCR

Total RNA was isolated from 5 × 10^5^ cells with the PerfectPure RNA Cultured Cell kit (5 Prime) and reverted with RevertAid reverse transcriptase (Thermo Scientific, Waltham, MA, USA) and random hexamer primers according to the manufacturer’s protocol. HCV replicon and β-actin used as a reference were detected by SYBR Green qPCR using a LightCycler^®^ 96 System (Roche). The primers for β-actin were: 5′-GATCATTGCTCCTCCTGAGC-3′ (sense), 5′-ACTCCTGCTTGCTGATCCAC-3′ (antisense); the primers for HCV (IRES) were: 5′-CTGTCTTCACGCAGAAAGCG-3′ (sense), 5′-GTCCTGGCAATTCCGGTGTA-3′ (antisense). Changes in the levels of each mRNA were evaluated by the ∆∆Ct method.

### 4.8. Western Blotting

Western blot analysis was performed principally according to an earlier published procedure [68]. Briefly, Huh7-luc/neo cells were seeded at a density of 1.4 × 10^5^ cells/well into 6-well culture plates (without antibiotic G418), cultured for 24 h (40–50% monolayer), and treated with the indicated concentrations of inhibitors for the specified time. After that, the medium was removed, and cells were washed with PBS and lysed with a lysis reagent (Promega, Madison, WI, USA) or RIPA lysis buffer (Thermo Scientific, Waltham, MA, USA). Proteins from supernatants were separated by electrophoresis in PAAG of a suitable percentage and electrotransferred onto a nitrocellulose membrane. The membrane was treated with 5% dry milk (Bio-Rad, USA) in PBST for 60 min at room temperature. Primary antibodies to NS3 (1:2000), NS5B (1:5000), LC3B (1:2000), SQSTM1/p62 (1:2000), acetylated lysine (1:1000), and α-tubulin (1:10,000) were added to the membrane, and it was incubated overnight at 4 °C and washed with PBST. A corresponding conjugate of horseradish peroxidase with secondary specific antibodies (1:10,000) was added to the membrane and it was incubated for 50 min at room temperature. Then, the membrane was washed with PBST, and the signal was visualized using an ECL Kit (Pierce Thermo Scientific, Waltham, MA, USA) and a High Performance ECL Film (GE Healthcare, Little Chalfont, UK).

### 4.9. Proteasome Activity Determination

The determination of proteasome activity in cells was performed as previously described [71]. Fluorescence was measured using a BD Fortessa Cytometer (BD Biosciences, San Jose, CA, USA). The data were analyzed with the ModFit LT software (Becton Dickinson, Franklin Lakes, NJ, USA).

### 4.10. DNA Content Analysis

The Huh7-luc/neo cells were seeded at a density of 1 × 10^5^ cells/well into 6-well culture plates (without antibiotic G418), cultured for 24 h, and treated with different concentrations of the test substances for an appropriate time. The cells were scraped off in 0.5 mL PBS and fixed by a gradual addition of 0.5 mL of cold ethanol with gentle vortexing. After incubation for 20 min, the cells were precipitated by centrifugation at 4000 rpm for 5 min. The supernatant was removed, and the cells were resuspended in 0.5 mL of PI-RNAse buffer (10 µg/mL PI and 0.1 mg/mL RNAse A in PBS) and incubated in the dark for 20 min. The DNA content in the cells was analyzed using an LSRFortessa Cytometer (BD Biosciences, San Jose, CA, USA). The data were analyzed with the ModFit LT software (Becton Dickinson, Franklin Lakes, NJ, USA).

### 4.11. Measurement of Reactive Oxygen Species

The production of intracellular reactive oxygen species (ROS) was measured in Huh7-luc/neo cells after 24 h of incubation with compounds. Then, medium was removed, and cells were incubated with 10 µM 2′,7′-dichlorofluorescein diacetate (*DCF-DA*) and washed 10 times with 500 µl of PBS. The fluorescence intensity (*DCF* intensity) was measured in 200 µl PBS using a Plate CHAMELEON V reader (Hidex, Turku, Finland) with excitation at 485 nm and emission at 535 nm.

### 4.12. SA-β-Galactosidase Staining of Cultured Cells

Huh7-luc/neo cells were incubated with palbociclib for 3 or 7 days, washed twice with PBS and fixed with 4% formaldehyde for 5 min, and stained with *X-Gal* as previously described [72]. After incubation overnight, the cells were washed with PBS twice and observed with a bright-field microscope at 10× magnification.

### 4.13. Establishing a Reporter Cell Line Stably Expressing LC3-GFP Using Lentiviral Transduction

HEK293T cells grown for 24 h in 35-mm Petri dishes to a confluence of 70–80% were transfected with 2.4 μg DNA (0.3 μg pLenti-CMV-GFP-LC3B-puro lentiviral vector and packaging plasmids: 0.6 μg pCMV-Rev, 1.2 μg pCMV-Gag, 0.3 μg pVSV-G) using 16 μL of TurboFect (Thermo Fisher Scientific, Waltham, MA, USA) according to the manufacturer’s instructions. After 4–8 h, the medium was replaced with 2–3 mL of the DMEM medium containing 2% FBS. The first stock of lentiviral particles was collected after 36 h, while the following stocks were collected every 24 h for 3–4 days, and frozen. After all stocks were collected, they were thawed, filtered through low-protein binding syringe filters (TPP) with pores of 0.45 μm, divided into 0.5 mL aliquots, and stored at −70 °C.

For lentiviral transduction, Huh7 cells were seeded into a 12-well plate and infected with a 0.5 mL aliquot of lentiviral particles at the confluence of 40–60%, and cells were seeded into a larger dish and selected with 1 μg/mL puromycin until uninfected control cells completely died (3–6 days).

### 4.14. Acquisition of Confocal Microscopy Images

Confocal images were acquired on a Leica DMI 6000B confocal microscope equipped with a Leica TCS SP5 laser scan unit (Leica Microsystems, Mannheim, Germany). Cells grown on coverslips were treated with substances and stained with a mix of dyes containing Hoechst 33342 (*Hoechst*), Lysotracker Red (*LTR*), and Bodipy 493/503 (*Bodipy*) to final concentrations of 5 µg/mL, 1 µM, and 0.5 µM, respectively, for another 30 min. The staining with Acridine Orange (*Acridine O.*) was always performed separately. Cells were stained with 26 µM *Acridine O.* for no more than 5 min. The cells stained with the dye mix without extra washing were examined under a microscope for 10–20 min, while the cells stained with *Acridine O.* were examined for 5–10 min. *Hoechst* was excited at 405 nm and detected in the 417–481 nm range; *LTR* was excited at 543 nm and detected in the 555–606 nm range; *Bodipy* was excited at 488 nm and detected in the 498–540 nm range; *Acridine O.* was excited at 458 nm and detected in the 467–549 nm and 584–680 nm ranges; *palbociclib* was excited at 405 nm and detected in the 582–633 nm range; *LC3-GFP* was excited at 488 nm and detected in the 498–540 nm range.

All images were acquired in “sequential scan mode” to completely avoid the “bleed-through” effect. Areas and intensities were measured using the ImageJ software (https://imagej.nih.gov/ij/, accessed on 21 January 2021). Colocalization analysis was performed using Coloc2 and JACoP [73] plugins. To validate colocalization, image set CBS001RGM-CBS010RGM from the Colocalization Benchmark Source (www.colocalization-benchmark.com, access on 21 January 2021) was used.

### 4.15. Statistical Data Analysis

All data were presented as the mean and standard deviation of at least three independent replicates. Statistical significance (assessed by the *t*-test) is shown in the figures: *** (*p* < 0.001), ** (0.001 < *p* < 0.01), * (0.01 < *p* < 0.05), # (0.05 < *p* < 0.1), ns—not significant.

## 5. Conclusions

This study shows that palbociclib, an inhibitor of cyclin-dependent kinases 4/6, induces in hepatoma cells a pre-senescent cellular phenotype and simultaneously accelerates viral replicon multiplication. This observation became a prerequisite for the creation of a test system for the screening of compounds under conditions characteristic of cellular senescence, which allowed us, for the first time, to suggest HDAC10 as a new potential target for anti-HCV treatment.

## Figures and Tables

**Figure 1 ijms-22-04559-f001:**
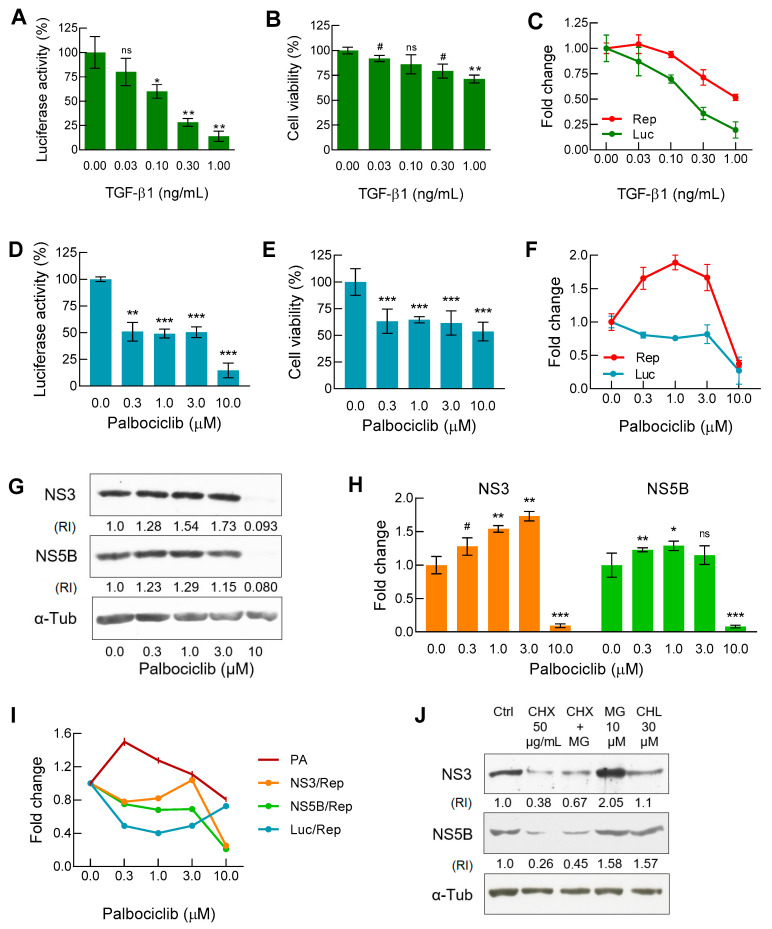
Testing of TGF-β1 and palbociclib in Huh7-luc/neo cell culture. (**A/D**) Luciferase activity (*n* = 3/4, respectively) and (**B/E**) cell viability (*n* = 3/6, respectively) were measured as described in Materials and Methods. (**C**/**F**) To assess the intracellular balance of luciferase (Luc) and viral replicon (Rep), the curves of normalized luciferase activity (La/Cv) and replicon content (HCV RNA/mRNA Actin) obtained by qRT-PCR (*n* = 3) were plotted. (**G**) Representative immunoblot and (**H**) graphical presentation of results of WB analysis of NS3 and NS5B content expressed as the relative intensity (RI) of either the NS3 or NS5B band versus the α-tubulin band (*n* = 3). (**I**) Dose–response curves for Luc, NS3, and NS5B normalized to values of replicon and proteasome activity (reference values of Luc and Rep correspond to data from Figure 1F, whereas the values of NS3 and NS5B correspond to data from Figure 1H). (**J**) Immunoblot analysis of NS3 and NS5B degradation pathways after treatment of cells for 18 h with cycloheximide (CHX), a translation inhibitor, MG132 (MG), a proteasomal activity inhibitor, and hydroxychloroquine (CHQ), a lysosomal acidic pH-neutralizing agent. In all cases, with the exception of Figure 1J, cells were treated with indicated concentrations of TGF-β1 and palbociclib for 48 h and 72 h, respectively; n is the number of independent experiments. Statistical significance (assessed by the *t*-test): *** (*p* < 0.001), ** (0.001 < *p* < 0.01), * (0.01 < *p* < 0.05), # (0.05 < *p* < 0.1), ns—not significant.

**Figure 2 ijms-22-04559-f002:**
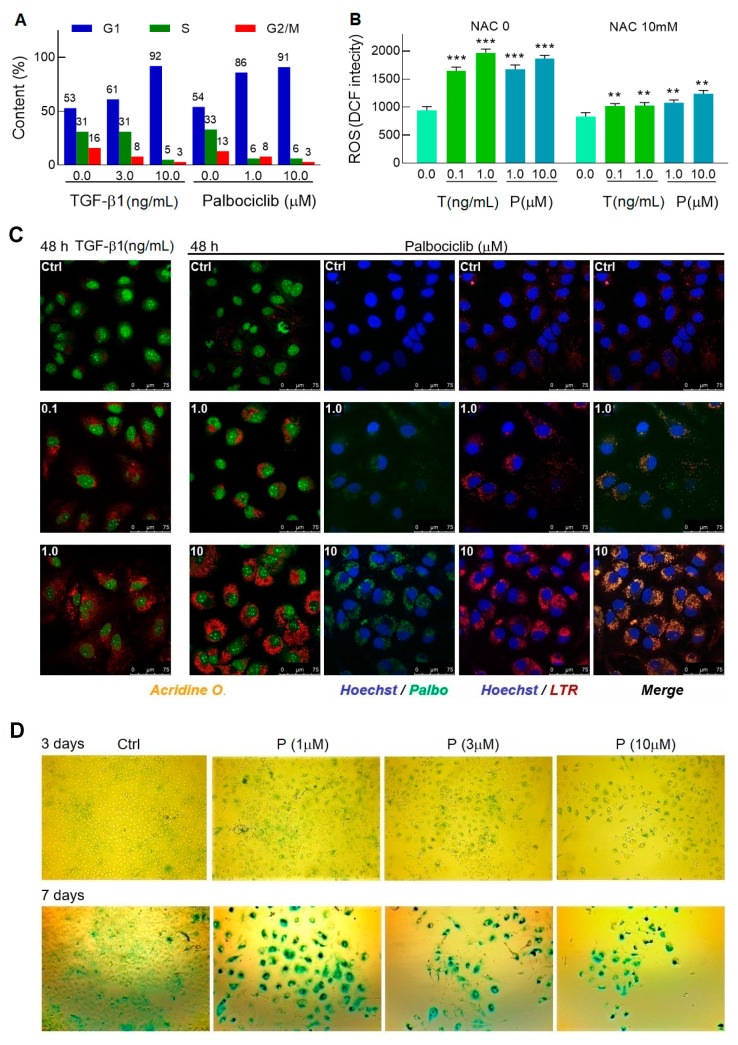
TGF-β1 and palbociclib launch the senescence program in Huh7-luc/neo cells. (**A**) Cells were treated with indicated concentrations of TGF-β1 and palbociclib for 48 h and 24 h, respectively, and analyzed for cell cycle distribution by flow cytometry. (**B**) 24 h of incubation with TGF-β1 and palbociclib induced oxidative stress that was manifested in ROS production detected by fluorometry of cells stained with *DCF-DA* (*n* = 5). (**C**) Confocal microscope images of *Acridine Orange*-stained cells treated with indicated concentrations of TGF-β1 and palbociclib (shown in the upper left corner) and colocalization of intrinsic fluorescence of *LTR* and palbociclib. (**D**) Light microscope images (×10 magnification) of *X-Gal*-stained cells treated with palbociclib for 3 and 7 days. Blue-green indicates positive staining for β-galactosidase activity; n is the number of independent experiments. Statistical significance (assessed by the *t*-test): *** (*p* < 0.001), ** (0.001 < *p* < 0.01).

**Figure 3 ijms-22-04559-f003:**
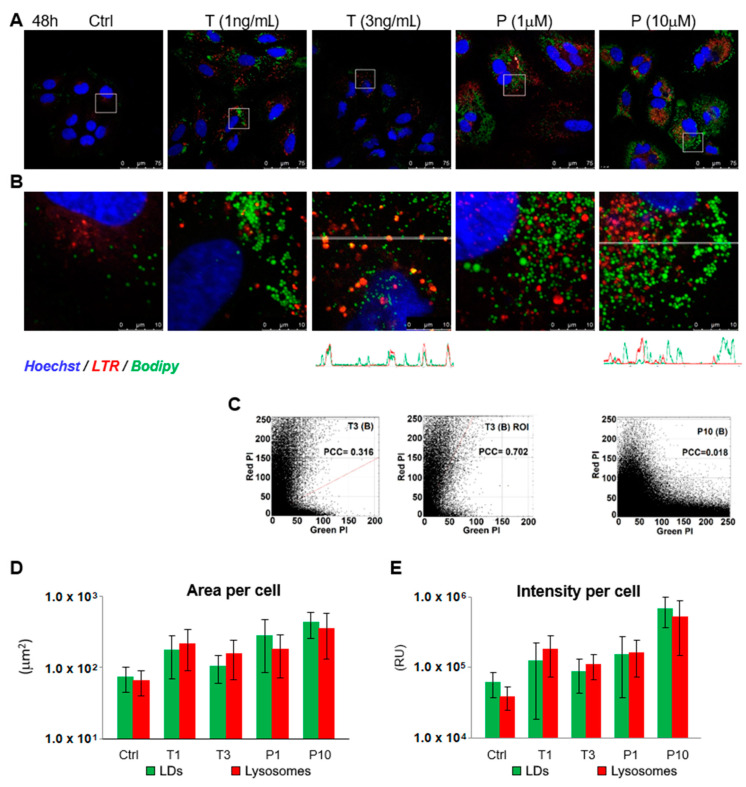
TGF-β1 and palbociclib cause accumulation of lipid droplets in hepatoma cells. (**A**,**B**) Confocal microscope images of *Hoechst-*, *LTR-*, and *Bodipy*-stained Huh7-luc/neo cells treated with indicated concentrations of TGF-β1 (T) and palbociclib (P) for 48 h and colocalization of *Bodipy* (LDs) and *LTR* (lysosomes) fluorescence signal indicated by red-green-blue (RGB) band profiles shown below the corresponding image. (**C**) Scatterplots of green and red pixel intensities (PI) of the cells shown in T3 (**B**) and P10 (**B**) and scatterplot in the region of interest (ROI) containing lysosomes in cells shown in T3 (**B**); PCC—Pearson’s correlation coefficient. (**D**) Area (µm^2^ per cell) occupied by green label (LDs) and red label (lysosomes) in cells shown in Figure 3A and (**E**) intensity of green (LDs) and red (lysosomes) fluorochromes; emission per cell was calculated as the sum of intensities (0–255) of all green and red pixels in a cell and expressed as relative units (RU). The area of 1 pixel is 0.0574 µm^2^.

**Figure 4 ijms-22-04559-f004:**
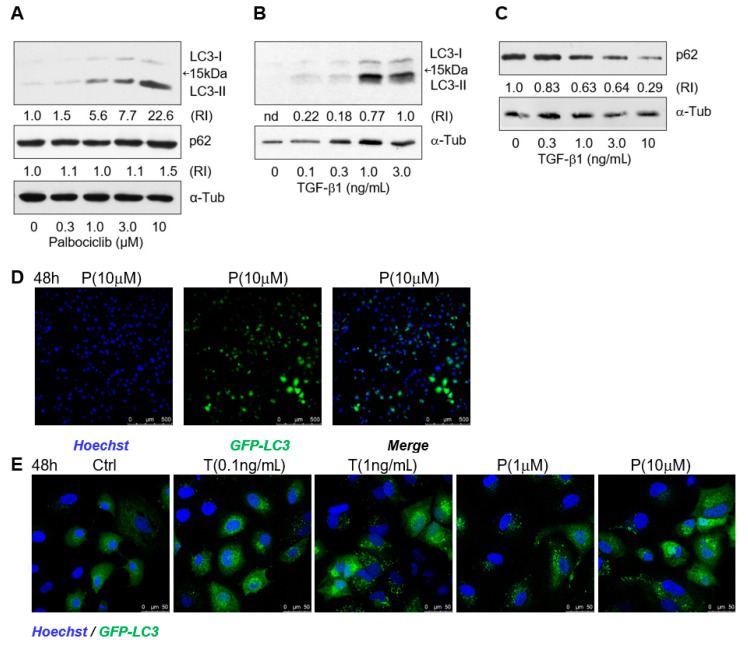
TGF-β1 and palbociclib activate biogenesis of autophagosome in hepatoma cells. (**A**) Western blots for LC3-I, LC3-II, and p62 in Huh7-luc/neo cells treated with palbociclib for 72 h and (**B**,**C**)with TGF-β1 for 48 h; α-tubulin was used as a loading control. Protein content is expressed as the relative intensity of either the LC3-II or p62 band versus the α-tubulin band. (**D**) Assessment of percentage of Huh7 cells stably expressing GFP-LC3 after 48 h of treatment of cells with 10 μM palbociclib (P). (**E**) Confocal microscope images of Huh7 cells stably expressing GFP-LC3 and treated with indicated concentrations of TGF-β1 (T) and palbociclib (P) for 48 h.

**Figure 5 ijms-22-04559-f005:**
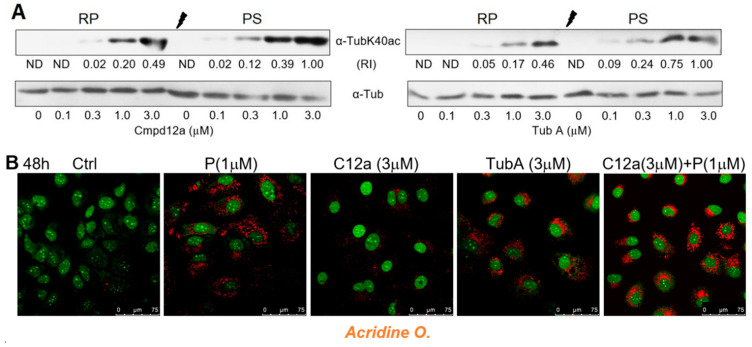
Inhibition of HDAC6 and HDAC6/10 by Cmpd12a and tubastatin A in Huh7-luc/neo cells. (**A**) Inhibition of HDAC6 deacetylation activity by Cmpd12a and tubastatin A in regularly proliferating (RP) and pre-senescent (PS) Huh7-luc/neo cells treated with indicated concentrations of inhibitors for 24 h, assessed by WB analysis of relative intensity (RI) of the α-tubulinK40ac band versus the α-tubulin band (ND—not defined). (**B**) Confocal microscope images of *Acridine Orange-*stained cells treated with indicated concentrations of Cmpd12a (C12a), tubastatin A (Tub A), and palbociclib (P) for 48 h.

**Table 1 ijms-22-04559-t001:** Testing of DAAs in regularly proliferating (RP) and in pre-senescent (PS) Huh7-luc/neo cells.

Inhibitor	Molecular Structure	HCV Target	Status of Cells	EC_50_ (µM)	CC_50_ (µM)	SI
				*p*		
Telaprevir	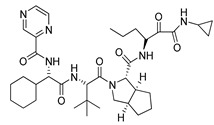	NS3 protease	RP	0.21 ± 0.02	100 ± 15	480
			PS	0.20 ± 0.05	230 ± 59	1150
				ns		
2′-Me-Ad	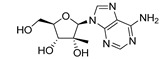	NS5B	RP	0.14 ± 0.01	140 ± 41	1000
			PS	0.12 ± 0.02	110 ± 5.0	920
				ns		
Sofosbuvir	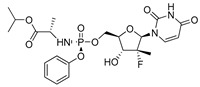	NS5B	RP	0.063 ± 0.002	>300	>4800
			PS	0.15 ± 0.03	>300	>2000
				**		
Ribavirin	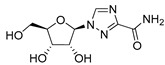	NS5B	RP	30 ± 5.0	>1000	>33
			PS	92 ± 8.0	>1000	>11
				***		

Cells were treated with different concentrations of the compounds for 48 h. Inhibitor concentrations causing 50% inhibition of HCV replication (EC_50_; *n* = 4) and 50% decline in cell viability (CC_50_; *n* = 6) were measured as described in Materials and Methods. The selectivity index (SI) represents the CC_50_/EC_50_ ratio. Statistical significance (assessed by the *t*-test): *** (*p* < 0.001), ** (0.001 < *p* < 0.01), ns—not significant.

**Table 2 ijms-22-04559-t002:** Testing of HTAs in regularly proliferating (RP) and in pre-senescent (PS) Huh7-luc/neo cells.

Inhibitor	Molecular Structure	Inhibition of HDACs	Status of Cells	EC_50_ (µM)	CC_50_ (µM)	SI
*p*
Vorinostat	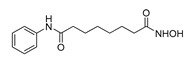	Class I and II	RP	0.62 ± 0.03	1.2 ± 0.22	1.9
PS	10 ± 1.8	>30	>3
***
Belinostat	* 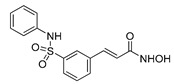 *	Class I and II	RP	0.23 ± 0.06	0.68 ± 0.02	3.0
PS	2.8 ± 0.25	>30	>11
***
CI-994	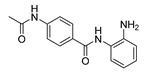	Class I 1/2/3	RP	8.3 ± 0.46	630 ± 35	76
PS	7.2 ± 1.9	480 ± 60	67
ns
SR-3212	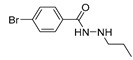	Class I 1/2/3	RP	1.3 ± 0.22	20 ± 1.6	15
PS	2.6 ± 0.72	>100	>39
*
*o*-PhO-CHA	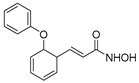	Class I 8	RP	0.56 ± 0.088	160 ± 10	290
PS	0.39 ± 0.065	170 ± 10	440
*
Cmpd13	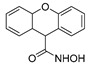	Class IIa 4/5/7	RP	0.85 ± 0.059	630 ± 40	740
PS	2.5 ± 0.57	760 ± 40	300
**
TMP-269	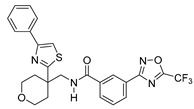	Class IIa 4/5/7/9	RP	5.1 ± 0.20	70 ± 7.0	14
PS	4.1 ± 0.20	>100	>24
***
Cmpd12a	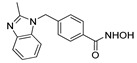	Class IIb 6	RP	0.270.08	13 ± 1.8	43
PS	2.8 ± 0.45	>30	>11
***
Tubastatin A	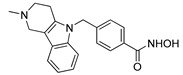	Class IIb 6/10	RP	1.0 ± 0.15	24 ± 2.8	20
PS	0.63 ± 0.11	>30	>48
*
Bufexamac	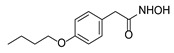	Class IIb 6/10	RP	5.3 ± 0.98	>300	>57
PS	4.3 ± 0.43	>300	>70
ns

Cells were treated with various concentrations of the compounds for 48 h. The concentrations of inhibitors causing a 50% inhibition of HCV replication (EC_50_; *n* = 4) and a 50% decline in cell viability (CC_50_; *n* = 6) were measured as described in Materials and Methods. The selectivity index (SI) represents the CC_50_/EC_50_ ratio. Statistical significance (assessed by the *t*-test): *** (*p* < 0.001), ** (0.001 < *p* < 0.01), * (0.01 < *p* < 0.05), ns—not significant.

## Data Availability

Not applicable.

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
