# Peer review of "Pre-Senescence Induction in Hepatoma Cells Favors Hepatitis C Virus Replication and Can Be Used in Exploring Antiviral Potential of Histone Deacetylase Inhibitors"

_ijms, 2021, doi:10.3390/ijms22094559_

Round 1

Reviewer 1 Report

The proposed manuscript investigated the ability of Palbociclib, a well-known inhibitor of cyclin-dependent kinases 4/6, to induce senescence in hepatocellular carcinoma Huh7-luc/neo cells, which are commonly used as an HCV replication platform to test the antiviral activity of potential inhibitor compounds. The ability of Palbociclib was also exploited to induce a pre-senescence cellular phenotype for studying how this state could affect both the HCV replicon and the inhibitory ability of some known anti-HCV agents (DAAs and HTAs). Finally, the authors proposed this model as a tool, to be used in combination with the standard replicon system, to discriminate test compounds that target/inhibit viral or host proteins strictly involved in the viral replication from antiproliferative agents. The proposed study would fit with the scope of the Journal and it would be of interest to researchers involved in anti-HCV drug discovery and beyond.

However, as it stands, the manuscript presents some serious concerns that should be addressed before it may be accepted for publication.

General comments:

- The authors did not provide enough evidence that the presence of 1µM Palbociclib has a null effect on the antiviral activity of the test compounds (or excluding any potential synergistic or antagonistic effects) and their cytotoxicity. In particular, as shown in figure 1B, 1µM Palbociclib resulted in a significant decrease of cell viability of almost 40-50% in MTT assays. The authors observed a constant effect over the range of 0.3-3 µM of Palbociclib in MTT as well as an accumulation of cells arrested in G1 phase, assuming that at those concentrations the drug caused only a cytostatic effect. However, a test to discriminate between cytostatic and cytotoxic effects seem to be required in order to confirm the absence of any toxic effect of 1µM Palbociclib on Huh7luc/neo cell lines.

- It is not clear or reported in the text how the authors calculated the EC50 and in particular CC50 values of compounds tested in the pre-senescence model inducted by 1µM Palbociclib. It is surprising to note that the CC50 values obtained by many compounds in the pre-senescent model were remarkably higher than in the standard Huh7-luc/neo cells.

Is there also any rationale why very different concentration ranges were tested in MTT for different test compounds (some compounds were tested up to 30 µM and others up to 300 or 1000 µM)?

- Overall, the validity and conclusion of the findings obtained testing the anti-HCV activities of compounds in the pre-senescent model seem to be quite vague and controversial and they require further explanation/evidence. Although the antiviral activity of vorinostat, belinostat and compound 12a decreased in pre-senescent cells, the selectivity index of vorinostat and especially belinostat significantly increased. Notably, the selectivity index of belinostat, defined by the authors as unsatisfactory, was comparable to that of ribavirin, also exhibiting a significant lower EC50 value. In addition, the cellular enzymes involved in Sofosbuvir activation (CatA and Hint1) and IMPDH, which is inhibited by ribavirin, are not dependent to the cell cycle progression. Therefore, they don’t seem to explain the reasons for their reduced anti-HCV activity in pre-senescent cells. Finally, the statistical significance of the difference of EC50 between two cellular conditions was not evaluated.

Minor comments:

- The resolution of confocal microscopic images is rather low, which makes them difficult to analyse.

Line 30 - The recent advance of HCV therapy due to the introduction of direct-acting antiviral (DAA) drugs/regimens should be mentioned in the introduction.

Line 62 – The authors should specify which control samples.

Line 92- TGF- β1 cannot be defined as a drug.

Line 203 – Figure 3D - The magnification of the control sample after 7 days is different to that of palbociclib-treated cells.

Line 280 - The definition of EC50, CC50 and TI (SI= selectivity index should be more appropriate) should be defined in the caption of Tables 1 and 2, together with the numbers of replicates for each experiment.

Line 200 – 202 – The number of cells seems to decrease over time in a dose-dependent manner. It is more reasonable to hypothesise a cytotoxic effect rather than a cytostatic one.

Line 207- Figure 4B- The magnification of the control and treated-cells seem to be different.

Line 263 – “with a more suitable habitat for HCV” should be replaced.

Line 293 – Table2 reports the same chemical structures for tubastin A and cmp12a.

Line 361- No data is available to assess and compare the reproducibility of the replicon assay in standard condition or in pre-senescence state.

Author Response

General comments:

Point 1:   The authors did not provide enough evidence that the presence of 1µM Palbociclib has a null effect on the antiviral activity of the test compounds (or excluding any potential synergistic or antagonistic effects) and their cytotoxicity.

Response 1:   We would like to note that the requested evidence that «the presence of 1µM Palbociclib has a null effect on the antiviral activity of the test compounds» has already been provided for DAAs – telaprevir and 2’-Me-Ad. Without a doubt, the authors agree with the reviewer that the antiviral effect of HDACi in the case of arrested test system may change, in particular, due to perturbations in the expression/activity of HDACs However, out of all selective HDACi, only Cmpd12a (inhibitor of HDAC6) showed a sharp drop in antiviral activity in senescent cells, but, as shown in paragraph 2.5, the HDAC6 activity in proliferating and arrested cells was almost equal.

Point 2:   In particular, as shown in figure 1B, 1µM Palbociclib resulted in a significant decrease of cell viability of almost 40-50% in MTT assays. The authors observed a constant effect over the range of 0.3-3 µM of Palbociclib in MTT as well as an accumulation of cells arrested in G1 phase, assuming that at those concentrations the drug caused only a cytostatic effect. However, a test to discriminate between cytostatic and cytotoxic effects seem to be required in order to confirm the absence of any toxic effect of 1µM Palbociclib on Huh7luc/neo cell lines.

Response 2:   The authors are grateful to the reviewer for these comments, the answers to which significantly helped to clarify the essence of the matter. Accordingly, we demonstrated the cytotoxic effect of palbociclib, which is reflected in the text of the article - "Despite the plateau of the MTT signal, the proportion of damaged cells measured by the values of propidium iodide uptake monotonically increased in comparison with the control from 9% to 19% with an increase in the concentration of palbociclib from 0.3 µM to 10 µM, respectively”.

Point 3:    It is not clear or reported in the text how the authors calculated the EC50 and in particular CC50 values of compounds tested in the pre-senescence model inducted by 1µM Palbociclib. It is surprising to note that the CC50 values obtained by many compounds in the pre-senescent model were remarkably higher than in the standard Huh7-luc/neo cells.

Response 3:   All the studied compounds have indeed a lower impact on the viability of cells previously treated with palbociclib (Tables 1 and 2). This can be explained by the fact that due to the G1-arrest in the cell cycle, the enzymes and factors responsible for genome duplication and cell division were absent in numerous cellular targets.

Poin 4:   Is there also any rationale why very different concentration ranges were tested in MTT for different test compounds (some compounds were tested up to 30 µM and others up to 300 or 1000 µM)?   

Response 4:    The authors agree with the reviewer that the concentration ranges of inhibitors in the MTT test look disparate, but they are often determined by different solubility of the compounds; in addition, in both test systems, the concentration range for the same inhibitor is always the same.

Point 5:   Overall, the validity and conclusion of the findings obtained testing the anti-HCV activities of compounds in the pre-senescent model seem to be quite vague and controversial and they require further explanation/evidence. Although the antiviral activity of vorinostat, belinostat and compound 12a decreased in pre-senescent cells, the selectivity index of vorinostat and especially belinostat significantly increased. Notably, the selectivity index of belinostat, defined by the authors as unsatisfactory, was comparable to that of ribavirin, also exhibiting a significant lower EC50 value.

Response 5:   In the case of belinostat, the authors primarily had in mind the high level of toxicity of this HDACi for the organism as a whole, judging by the results of its clinical trials as an anticancer agent (FDA approval: belinostat for the treatment of patients with relapsed or refractory peripheral p-cell lymphoma. doi:10.1158/1078-0432.CCR-14-3119). Based on this, the authors concluded that an increase in the ЕС50 value up to 10 μM in senescent cells, which by 8 times exceeds the value of СС50=1.2 µM in proliferating cells, will make the use of belinostat for the treatment of senescent HCV-infected hepatocytes unacceptable due to the high likelihood of side effects.

Point 6:   In addition, the cellular enzymes involved in Sofosbuvir activation (CatA and Hint1) and IMPDH, which is inhibited by ribavirin, are not dependent to the cell cycle progression. Therefore, they don’t seem to explain the reasons for their reduced anti-HCV activity in pre-senescent cells.

Response 6:  Obviously, this issue requires more detailed study, which is in our immediate plans. However, the effect of palbociclib is not limited to the G1 arrest in the cell cycle. According to our and literature data, it cannot be ruled out that the effect of palbociclib on proteasome activity leads to degradation of IMPDH (ANKRD9 is associated with tumor suppression as a substrate receptor subunit of ubiquitin ligase. https://doi.org/10.1016/j.bbadis.2018.07.001).

Point 7:   Finally, the statistical significance of the difference of EC50 between two cellular conditions was not evaluated.

Response 7:  In the new version, the statistical significance of the difference of EC50 between two cellular conditions was entered into both tables.

Minor comments:

Point 8:   The resolution of confocal microscopic images is rather low, which makes them difficult to analyse.

Response 8:   The resolution of the original confocal microscopic images was much better in the first version of the manuscript and was probably reduced due to image conversion during editorial changes to the manuscript. In case of recurrence, we plan to resolve this problem with the technical editor of the journal.

Point 9:   Line 30 - The recent advance of HCV therapy due to the introduction of direct-acting antiviral (DAA) drugs/regimens should be mentioned in the introduction.

Response 9:   The introduction now begins with a sentence: “The success achieved in the modern therapy of hepatitis C due to the use of direct acting antivirals (DAAs) gives time to find new approaches to the treatment of this disease.”

Point 10:   Line 62 – The authors should specify which control samples.

Response 10:   In the case of experiments with TGF-b1 and palbociclib, an appropriate volume of PBS was added to the control samples, since both substances were prepared and added to the wells as solutions in PBS. Control samples in experiments with antiviral agents were treated with 1% v/v of ethanol. Cells treatment protocol: the stock solution of the compound (usually 100 mM in DMSO) was serially diluted with alcohol to 100x concentration; 1% of the total volume was added to each well – 5 μl in case of a 48-well plate and 2 μl in case of a 96-well plate.

Point 11:   Line 92 - TGF- β1 cannot be defined as a drug.

Response 11:   TGF-β1 is defined in the new version as a substance.

Point 12:   Line 203 – Figure 3D - The magnification of the control sample after 7 days is different to that of palbociclib-treated cells.

Response 12:    It may appear so, but in fact the magnification is exactly the same. The increase in cell size can be explained by the development of cellular senescence.

Point 13:   Line 280 - The definition of EC50, CC50 and TI (SI= selectivity index should be more appropriate) should be defined in the caption of Tables 1 and 2, together with the numbers of replicates for each experiment.

Response 13:   Everything has been done. The captions of Tables 1 and 2 now contain the definitions: “Inhibitor concentrations causing 50% inhibition of HCV replication (EC50; n=4) and 50% decline in cell viability (CC50; n=6) were measured as described in the Materials and Methods. The selectivity index (SI) represents the CC50/EC50 ratio. Statistical significance (assessed by the t-test): *** (p < 0.001), ** (0.001 < p < 0.01), * (0.01 < p < 0.05), # (0.05 < p < 0.1), ns – not significant.”

Point 14:   Line 200 – 202 – The number of cells seems to decrease over time in a dose-dependent manner. It is more reasonable to hypothesise a cytotoxic effect rather than a cytostatic one.

Response 14:   As noted above, we determined the proportion of cells with damaged membrane in the presence of palbociclib in the concentration range of 0.3-10 μM. To accommodate this reviewer’s comment, the authors added a correction: «…clearly illustrating the strong cytostatic as well as cytotoxic effects of the drug».

Point 15:   Line 207- Figure 4B- The magnification of the control and treated-cells seem to be different.

Response 15:   This may be a mistake of perception. The magnification was exactly the same.

Point 16:   Line 263 – “with a more suitable habitat for HCV” should be replaced.

Response 16:   It was replaced with “for screening of compounds under conditions characteristic of cellular senescence.”

Point 17:   Line 293 – Table2 reports the same chemical structures for tubastin A and cmp12a/

Response 17:   The molecular structure of tubastatin A was corrected.

Point 18:   Line 361- No data is available to assess and compare the reproducibility of the replicon assay in standard condition or in pre-senescence state.

Response 18:   The corresponding sentence “Furthermore, this approach ensured good repeatability and reproducibility of measurements comparable to those in the regularly proliferating cells.” was deleted as being superfluous.

Reviewer 2 Report

In the manuscript “Pre-Senescence Induction in Hepatoma Cells Favors Hepatitis 
3C Virus Replication and Can Be Used in Exploring Antiviral 4 Potential of Host-Targeting Agents”, Malikova and colleagues report the development of a protocol to induce pre-senescence in Hepatoma cells harboring an HCV subgenomic replicon, and use such system to evaluate pre-sensescence’s effect on both HCV replication and sensitivity to drugs. The author’s hypothesis is that drug potency tested on in vitro cultured Hepatoma cells is poorly indicative of drug properties in vivo, since HCV infection induces significantly changes in hepatocytes properties during chronic infection – including senescence. Therefore, they decide to develop a pre-senescent Hepatoma cell to more physiologically evaluate the effects of certain drugs, with a specific focus on host-directed targets. They find that palbociclib treatment can indeed induce senescence, which is also linked to an increase of LD, lysosomes and, up to drug concentrations below 10 uM, HCV replication. The authors their system to study the effect senescence on cytotoxicity and antiviral activity of several DAAs (including protease and polymerase inhibitors but not NS5A inhibitors) on a few host targeting agents (unfortunately including only HDAC inhibitors – but not Cyclophilin, mir-122 or PI4KIIIa inhibitors). The authors find unsurprisingly that the behavior of most DAA is unaffected (with the notable exception of Sofosbuvir, which however is a prodrug), while CC50s of HDACi are greatly increased, as expected by cells which are cell cycle arrested.

The system develop by the authors is intriguing, the manuscript is clearly written, data are logically presented, and results are interesting. However, the work is intimately endowed with a number of important limitations of diverse nature, including “philosophical”, technical and logical ones. Nevertheless, the manuscript as good potential and I would therefore strongly encourage resubmission after addressing the following concerns.

Main points

  • Title: “Exploring Antiviral Potential of Host-Targeting Agents”. The title seems to suggest that the main focus of the manuscript is to develop a better system to screen new HCV antivirals targeting host factors. This is quite confusing since the approval of DAA has almost completely abolished the need to develop news antiviral drugs, such that all companies stopped any drug discovery activity, including for very advanced and promising drugs. However, it is impossible to predict what will happen in the future, and research should be free, so I would not like to further comment on this limitation, beside suggesting to slightly expanding the sentence “to combat this pathogen and may solve the problem of viral resistance that is typically associated with the treatment by direct-acting antivirals (DAAs)“. On the other hand, however, the authors fail to address the issue they raised (i.e. testing HTA in their system) since the present version of the manuscript only focuses on a specific class of HTA: HDAC inhibitors. This reviewer is aware that the authors have a long-lasting experience in the field of HDACi as HCV antivirals, but additional, more established HTAs have to be included, such as Alisporivir and Marivirsen. Alternatively, the authors should restructure the manuscript to focus on the utility of their system to study the antiviral effect of HDACi. Indeed, HADCi are also used as anticancer compounds, and therefore would kill Hepatoma cells, making the latter an unsuitable system. In this context, the manuscript would be much more to the point. Accordingly, induction of pre-senescence greatly increased the CC50 of all HDACi tested (see Table 2).
  • INTRODUCTION: “To improve the efficiency of HTA screening, the functional status of hepatoma cells in the test system must be obviously brought closer to the status of chronically HCV-infected hepatocytes.” It is not clear how inducing pre-senescence would render Hepatoma cell similar to a chronically infected hepatocyte. Indeed, there are a few reports (properly cited, actually, although I would suggest including the work of Valérie Paradis published in Human Pathology in 2001) indicating that liver cells undergo senescence, but this appears to be strongly linked to progression of viral infection and liver damage, and therefore senescence would appear to be a characteristic feature of long-lasting chronic infection, where hepatocytes already experienced functional disruption. This should be clarified at lines 51-53 by expanding current wording. Furthermore, the authors must stress that Hepatoma cells are still very far away from a physiological condition which would require a proper liver architecture and primary untransformed cells. The latter is obviously an extremely difficult task to accomplish, but such limitation is worth being mentioned, as well as a comparison with other methods previously developed that would also more closely mirror more physiological HCV culture conditions including, but not limited to: DMSO arrested cultures (as shown by Bruno Sainz in 2006 – a system which particularly worth being discussed given the growth arrested phenotype of cells) and Matrigel-embedded 3D cultures (as shown by Francisca Molina-Jimenez in 2012).

  • The experimental protocol to induce pre-senescence appears to be linked to very high cytotoxicity and therefore results are of dubious interpretation. In Figure 1E clearly shows a very strong (40%) reduction in cell metabolic activity after 48-72h treatment (actual duration of treatment is not clear). The fact that 2x10^4 were seeded in a 96 well plate implies that cells are almost – if not completely – confluent at the time compounds are added, and hence the phenotype cannot entirely attributable to a delay in cell division (as expected from compounds inhibiting cdk activity) but rather more profound changes in metabolism and viability. The authors need to further investigate the reason behind such phenotype and quantify cell death (apoptosis and necrosis). Furthermore, additional assays must be performed varying number of cell seeded (starting from example from 2x10^2 cells/well) and duration of treatment (for example 6, 12, 24 and 48 hours). Additionally I would recommend also testing for ATP intracellular content (for example using cell titer glow® assay from Promega) in order to provide an alternative readout to MTT assays. These are crucial points since it is hard to really consider results reliable when dealing with a 40% in MTT assays upon treatment.

  • All experiments are performed in Huh7 – replicon cell lines and not on the control Huh7 cell line alone. This makes the reader wonder if what the authors observe in terms of pre-senescence, lipid metabolism, an autophagy is a HCV related phenotype or not. Clearly cell viability experiments in Figure 1 and all experiments in Figure 2 should be performed in Huh7 cells. The same holds true for the important rise in CC50 concentration observed upon treatment of pre-senescence cells with HDACi from Table 2.

Minor issues

  • Please expand the section dealing with HTAs and mention why the authors believe HDACi are a promising class of agents, and explain the main limitation to Huh7 cells to study their activity.
  • All images are of very low resolution and are therefore very difficult to be properly evaluated.
  • Throughout the manuscript: authors refer to untreated cells as to “regularly proliferating”: given the fact that they are probably overconfluent at the time of analysis, in the absence of a FACS-based cell cycle analysis, I cannot be sure of such definition. In the absence of such experimental evidence, cells should be referred as “untreated”
  • All experiments. How are “untreated” cells treated? Does it involve ethanol or DMSO treatment? At which concentration. Please indicate such information or repeat experiments including appropriate vehicle treated cells.
  • Figure 1. The increase of HCV replication observed in panel F upon palbociclib should be discussed on the light of evidence that cell overconfluency results in a decrease of HCV subgenomic replicon replication (see Hossein M Elbadawy,2020). Indeed, the evidence that cell confluency influences HCV replication should be taken into account when interpreting such data. Given the experimental protocol followed by the authors, whereby 4x10^4 cells are seeded in a 24 well plate, let grow for 24h, then treated with palbociclib, further incubated for 72h, it is very likely that untreated cells would be way overconfluent, while the palbociclib would not as a consequence of cell-cycle arrest. In other words, is the increase in relative HCV replicaton due to reduced cell overconfluency? In addition to comment on this I would suggest the authors to try seeding different cell numbers and annotate cell confluency for each experiment involving viral replication quantification.
  • Figure 3. Please provide quantification of Lipid droplets and Lysosomes number and intensity as well as LDs and Ls co-localization (Pearson).
  • Figure 3 and discussion. The authors imply several times that Pal treatment results in an alteration of authophagosome/lysosome/LDs pathway. This mainly stems from co-localization between Bodipy and lysosome markers, but in the absence of properly controlled experiments, no conclusion can actually be drawn. This could indeed be a very intriguing finding is further supported by experimental evidence.
  • Table 2. The authors should clarify why they only tested HDACi as HTAs.
  • Table 2: Apparently all compounds have already been tested for CC50 and ED50 vs. HCV. This should be made obvious in the text.
  • Table 2: Something might be wrong in the chemical structures of Compound 12 and Tubasmatin, since they appear identical to this reviewer. Please double check.
  • Table 2 and throughout the manuscript. In a previous report by the same group, Tubasmatin was used as a HDAC 6 specific inhibitor, now it has been “promoted” to HDAC 6 and 10 inhibitor. This should be clarified. Indeed, it is commonly known as a highly HDAC 6 inhibitor, although it can inhibit to a lesser extend also HDAC8.
  • Line 328 please provide an appropriate reference here.

Author Response

Main points

Point 1:   Title: “Exploring Antiviral Potential of Host-Targeting Agents”. The title seems to suggest that the main focus of the manuscript is to develop a better system to screen new HCV antivirals targeting host factors. This is quite confusing since the approval of DAA has almost completely abolished the need to develop news antiviral drugs, such that all companies stopped any drug discovery activity, including for very advanced and promising drugs.

Response 1:    This part of the title was changed to “Exploring Antiviral Potential of Histone Deacetylase Inhibitors”.

Point 2:   However, it is impossible to predict what will happen in the future, and research should be free, so I would not like to further comment on this limitation, beside suggesting to slightly expanding the sentence “to combat this pathogen and may solve the problem of viral resistance that is typically associated with the treatment by direct-acting antivirals (DAAs)“

Response 2:   The sentence was accordingly changed to “to combat this pathogen and address the challenges that lay ahead commonly associated with the viral resistance to treatment by DAAs“.

Point 3:   On the other hand, however, the authors fail to address the issue they raised (i.e. testing HTA in their system) since the present version of the manuscript only focuses on a specific class of HTA: HDAC inhibitors. This reviewer is aware that the authors have a long-lasting experience in the field of HDACi as HCV antivirals, but additional, more established HTAs have to be included, such as Alisporivir and Marivirsen. Alternatively, the authors should restructure the manuscript to focus on the utility of their system to study the antiviral effect of HDACi. Indeed, HADCi are also used as anticancer compounds, and therefore would kill Hepatoma cells, making the latter an unsuitable system. In this context, the manuscript would be much more to the point. Accordingly, induction of pre-senescence greatly increased the CC50 of all HDACi tested (see Table 2).

Response 3:   It was done. The number of HDACi was increased from 5 to 10, so that the selected compounds cover a wide spectrum of HDACi inhibition selectivity.

Point 4:   INTRODUCTION: “To improve the efficiency of HTA screening, the functional status of hepatoma cells in the test system must be obviously brought closer to the status of chronically HCV-infected hepatocytes.” It is not clear how inducing pre-senescence would render Hepatoma cell similar to a chronically infected hepatocyte.

Response 4:    With this particular phrase, the authors wanted to draw attention to the importance of testing HTAs in systems that specifically simulate a chronic infection.

Point 5:   Indeed, there are a few reports (properly cited, actually, although I would suggest including the work of Valérie Paradis published in Human Pathology in 2001) indicating that liver cells undergo senescence.

Response 5:    It was done as reference [22].

Point 6:   but this appears to be strongly linked to progression of viral infection and liver damage, and therefore senescence would appear to be a characteristic feature of long-lasting chronic infection, where hepatocytes already experienced functional disruption.

Response 6:   As far as we know, it has not yet been described in the literature how quickly the presence of the virus induces cellular senescence and to what extent it correlates with the functional disruption of the host hepatocyte. In addition to direct viral effect, cellular senescence can be induced by the paracrine action of SASP, and cells thus aged can also be attacked by HCV. The approach that we use (of course, only as the very first approximation) simulates “senescence-induced senescence” and makes it possible to study how cellular senescence affects HCV replication.

Point 7:   This should be clarified at lines 51-53 by expanding current wording.

Response 7:   We made an insertion into the text: “In addition, senescent hepatocytes can accumulate in damaged liver tissues as a result of a process known as “senescence-induced senescence” [25].    

Point 8:   Furthermore, the authors must stress that Hepatoma cells are still very far away from a physiological condition which would require a proper liver architecture and primary untransformed cells. The latter is obviously an extremely difficult task to accomplish, but such limitation is worth being mentioned, as well as a comparison with other methods previously developed that would also more closely mirror more physiological HCV culture conditions including, but not limited to: DMSO arrested cultures (as shown by Bruno Sainz in 2006 – a system which particularly worth being discussed given the growth arrested phenotype of cells) and Matrigel-embedded 3D cultures (as shown by Francisca Molina-Jimenez in 2012).

Response 8:   For the sake of rigor, in the new version of the article the authors do not use the expression “a test system with a more suitable “habitat” for HCV” and in general do not carry out any comparison of pre-senescent hepatoma cells with chronically infected hepatocytes or primary untransformed cells. The aim of the work is formulated differently, namely - “In the present work, we explored the feasibility of using palbociclib as a low molecular weight analogue of TGF-β1 to induce senescence in hepatoma cells and to study how pre-senescent status of the cells affects HCV replication and the efficacy of representative anti-HCV agents”.

Point 9:   The experimental protocol to induce pre-senescence appears to be linked to very high cytotoxicity and therefore results are of dubious interpretation. In Figure 1E clearly shows a very strong (40%) reduction in cell metabolic activity after 48-72h treatment (actual duration of treatment is not clear).

Response 9:   At the end of the caption to Figure 1, there is an explanation – “In all cases, with the exception of Figure 1J, cells were treated with indicated concentrations of TGF-β1 and palbociclib for 48h and 72h, correspondingly”. Indeed, the authors received confirmation of the remark of the reviewer (which will be discussed later) that the reduction in the signal in MTT tests is due to the combined cytotoxic and cytostatic effects of palbociclib.  

Point 10:   The fact that 2x10^4 were seeded in a 96 well plate implies that cells are almost – if not completely – confluent at the time compounds are added, and hence the phenotype cannot entirely attributable to a delay in cell division (as expected from compounds inhibiting cdk activity) but rather more profound changes in metabolism and viability.

Response 10:    With gratitude to the reviewer, the authors acknowledge their negligence in the provided values of the number of cells. The fact is that the values of 8x104 and 2x104 cells/well well for 24 and 96-well plates, respectively, were erroneously reported in the article by Kozlov et al., 2015 (doi:10.1016/j.bmcl.2015.04.016) and were uncritically transferred to this article. We rechecked the values using a Goryaev chamber and found that the number of adhered cells in 48 and 96-well plates 24 hours after seeding was in fact about 1x104 and 3.5x103 cells/well, respectively, with a cell confluence of 30-40% that only after 2-3 days of incubation reached 90-100% in the control wells, which is clearly seen on the light microscope image (Figure 2D).    

Point 11:   The authors need to further investigate the reason behind such phenotype and quantify cell death (apoptosis and necrosis). Furthermore, additional assays must be performed varying number of cell seeded (starting from example from 2x10^2 cells/well) and duration of treatment (for example 6, 12, 24 and 48 hours). Additionally I would recommend also testing for ATP intracellular content (for example using cell titer glow® assay from Promega) in order to provide an alternative readout to MTT assays. These are crucial points since it is hard to really consider results reliable when dealing with a 40% in MTT assays upon treatment.

Response 11:   At the request of the reviewer, we carried out additional experiments and demonstrated the presence of the cytotoxic effect of palbociclib, which was reflected in the text of the article - “Despite the plateau of the MTT signal, the proportion of damaged cells measured by the values of propidium iodide uptake monotonically increased in comparison with the control from 9% to 19% with an increase in the concentration of palbociclib from 0.3 µM to 10 µM, respectively ". In view of the above, the authors hope that they were able to dispel the reviewer's reservations about the correct interpretation of the results presented in the article.

Minor issues

Point 12:   Please expand the section dealing with HTAs and mention why the authors believe HDACi are a promising class of agents, and explain the main limitation to Huh7 cells to study their activity.

Response 12:   The corresponding paragraph was added - “HDACi is a promising class of anti-HCV agents, some of which inhibit the multiplication of HCV in vitro and in vivo [50,51], while others slow down the development of hepatocellular carcinoma [52]. Potentially, such a combination of activities in one compound may provide an advantage in the treatment of late stages of CHC accompanied by carcinogenesis [53], but, on the other hand, it also complicates HDACi testing in regularly proliferating hepatoma cells.”

Point 13:   All images are of very low resolution and are therefore very difficult to be properly evaluated.

Response 13:       The resolution of the original confocal microscopic images was much better in the first version of the manuscript and was probably reduced due to image conversion during editorial changes to the manuscript. In case of recurrence, we plan to resolve this problem with the technical editor of the journal.

Point 14:   Throughout the manuscript: authors refer to untreated cells as to “regularly proliferating”: given the fact that they are probably overconfluent at the time of analysis, in the absence of a FACS-based cell cycle analysis, I cannot be sure of such definition. In the absence of such experimental evidence, cells should be referred as “untreated”.

Response 14:   The problem of cell overconfluence is a consequence of an erroneous overestimation of the number of cells in the wells of the plates that we acknowledged above; in reality, there is no such problem. In all experiments to establish the values of ЕС50 and СС50, the confluence of cells 24 hours after seeding was 30-40% and only after 2-3 days of incubation it reached 90-100% in the control wells, as shown on the light microscope image (Figure 2D).

Point 15:   All experiments. How are “untreated” cells treated? Does it involve ethanol or DMSO treatment? At which concentration. Please indicate such information or repeat experiments including appropriate vehicle treated cells.

Response 15:    Cell treatment protocol: the stock solution of the compound (usually 100 mM in DMSO) was serially diluted with alcohol to 100x concentration; 1% of the total volume was added to each well – 5 μl in case of a 48-well plate and 2 μl in case of a 96-well plate. Control samples in experiments with antiviral agents were treated with 1% v/v of ethanol. In the case of experiments with TGF-b1 and palbociclib, an appropriate volume of PBS was added to the control samples, since both substances were prepared and added to the wells as solutions in PBS.

Point 16:   Figure 1. The increase of HCV replication observed in panel F upon palbociclib should be discussed on the light of evidence that cell overconfluency results in a decrease of HCV subgenomic replicon replication (see Hossein M Elbadawy,2020). Indeed, the evidence that cell confluency influences HCV replication should be taken into account when interpreting such data. Given the experimental protocol followed by the authors, whereby 4x10^4 cells are seeded in a 24 well plate, let grow for 24h, then treated with palbociclib, further incubated for 72h, it is very likely that untreated cells would be way overconfluent, while the palbociclib would not as a consequence of cell-cycle arrest. In other words, is the increase in relative HCV replicaton due to reduced cell overconfluency? In addition to comment on this I would suggest the authors to try seeding different cell numbers and annotate cell confluency for each experiment involving viral replication quantification.

Response 16:   As mentioned above, the problem of cell overconfluence does not exist - the authors erroneously overestimated the number of cells in the wells of the plates. In fact, in all experiments to establish the values of EC50 and CC50, the confluence of cells 24 hours after seeding was 30-40% and only after 2-3 days of incubation it reached 90-100% in the control wells, as shown on the light microscope image (Figure 2D).

Point 17:   Figure 3. Please provide quantification of Lipid droplets and Lysosomes number and intensity as well as LDs and Ls co-localization (Pearson).

Response 17:    It is difficult to count the number of organelles because their images on cell sections overlap. In addition, there are dense clusters of the organelles. Therefore, in the new version, we measured the average area occupied by these organelles on the optical section of one cell (Figure 3A). Colocalization analysis was performed for two out of five parts of Figure 3В using Coloc2 and JACoP plugins. Image set CBS001RGM-CBS010RGM from the Colocalization Benchmark Source (www.colocalization-benchmark.com) was used to validate colocalization. Figure 3C presents the corresponding 2D cytofluorograms with values of Pearson’s coefficients. Corresponding changes were made to Materials and Methods.

Point 18:   Figure 3 and discussion. The authors imply several times that Pal treatment results in an alteration of authophagosome/lysosome/LDs pathway. This mainly stems from co-localization between Bodipy and lysosome markers, but in the absence of properly controlled experiments, no conclusion can actually be drawn. This could indeed be a very intriguing finding is further supported by experimental evidence.

Response 18:    The authors fully agree with the reviewer that additional argumentation on this important issue was needed. In the new version, we present the results of the Western blot analysis of the LC3-II and p62 content in the Huh7-luc/neo hepatoma cells after their incubation with TGF-b1 (Figure 4A/B). The detected increase in the expression of LC3-II with a simultaneous decrease in the p62 content is characteristic of the normal course of autophagy and it sharply contrasts with the accumulation of p62 in the presence of palbociclib. The corresponding remark was made in paragraph 2.3. - “On the contrary, in the presence of TGF-β1 in the concentration range of 0.3-3ng/ml, the p62 level decreased with a simultaneous increase in the LC3-II content (Figure 4A/B), which is in good agreement with the active course of lipophagy that was also identified by microscopy (Figure 3B)", and in the Discussion – “It is worth noting that the action of palbociclib in this aspect is very similar to autophagy inhibition by chloroquine [64]. This is probably a consequence of the similarity of the molecular structure of both compounds, where the heterocyclic nucleus and the alkylammonium residue are linked together through a short linker.”  

Point 19:   Table 2. The authors should clarify why they only tested HDACi as HTAs.

Response 19:    The corresponding paragraph was added – “HDACi is a promising class of anti-HCV agents etc.” See our reply above.

Point 20:   Table 2: Apparently all compounds have already been tested for CC50 and ED50 vs. HCV. This should be made obvious in the text.

Response 20:    For a correct comparison of test results in two systems, the time of incubation with anti-virus agents was changed from 72 hours to 48 hours.

Point 21:   Table 2: Something might be wrong in the chemical structures of Compound 12 and Tubastatin A, since they appear identical to this reviewer. Please double check.

Response 21:    The structure of Tubastatin A was corrected.

Point 22:   Table 2 and throughout the manuscript. In a previous report by the same group, Tubastatin was used as a HDAC 6 specific inhibitor, now it has been “promoted” to HDAC 6 and 10 inhibitor. This should be clarified. Indeed, it is commonly known as a highly HDAC 6 inhibitor, although it can inhibit to a lesser extend also HDAC8.

Response 22:    We added the following clarification into the Discussion - “It has recently been shown that the tubastatin A scaffold ensures simultaneous binding to HDAC6 and HDAC10.”

Point 23:   Line 328 please provide an appropriate reference here. (development of infection and its treatment at the most severe stages of the disease).

Response 23: The corresponding reference [60] to work Mucke2017 was made in this context.  

Round 2

Reviewer 1 Report

The authors have addressed some points raised and improved the quality of their work. However, for other points, they made efforts to reply to the reviewer but did not take any actions to amend the text.

 This paper will be suitable to be published in International Journal of Molecular Sciences, after addressing the following minor points.

 - As stated in the revised text, the authors measured the cytotoxicity of palbociclib by propidium iodide uptake, but no data is shown. The related results should be reported in the manuscript or optionally in supplementary info. Besides, the narrative text of such a result should also be clearer or more appropriate (lines109-112), as it is not well linked to the previous sentence. This test was necessary to discriminate the cytotoxic/cytostatic effects observed in MTT assay.

- The authors should report in the Material and Methods sections how they calculate the EC50 and CC50 values in the pre-senescence model (in the presence of 1μM palbociclib)

- The authors reply regarding belinostat/vorinostat is still uncertain (obviously their anticancer effects is well known). In particular, it is not clear why they compared their antiviral activity in the pre-senescent cells with the cytotoxicity observed in regularly proliferating cells. I suggest revising the text (lines 509-515) with a focus on the reduced potency of their antiviral activity rather than their therapeutic potential.

- Title of Section 2.4 should be revised, as “Testing antiviral drugs under regular and pre-senescence conditions allows to distinguish between DAAs and HTAs” is not supported by the presented results in this manuscript.

- Surprisingly, the p-value (< 0.001) reported in Table2 indicates a significant difference in the TMP-269 antiviral activity in both cellular conditions, however in the text, it is reported that no difference was observed with this compound. Please revise it.

- Lines 33-35. The sentence introduced does not make sense. Please revise it.

- Line 72 (I simply meant to define the kind of controls in the sentence reported in line 72, “…had shown a sharp increase in the concentration of TGF-β1 in periportal hepatocytes and hepatic sinusoidal cells in comparison with control samples”) – Please define which kind of control (from healthy individuals?). Line 125 please refer to untreated control or solvent/vehicle control.

-please thoroughly revise English and technical language as some mistakes have been detected throughout the text. Revise also some sentences (e.g. lines 121-122).

Author Response

Point 1:   As stated in the revised text, the authors measured the cytotoxicity of palbociclib by propidium iodide uptake, but no data is shown. The related results should be reported in the manuscript or optionally in supplementary info. Besides, the narrative text of such a result should also be clearer or more appropriate (lines109-112), as it is not well linked to the previous sentence. This test was necessary to discriminate the cytotoxic/cytostatic effects observed in MTT assay.

Response 1:   The text fragment indicated by the Reviewer was modified as follows – “To discriminate the cytotoxic/cytostatic effects observed in MTT assay, the proportion of damaged cells was assessed by the values of propidium iodide uptake [34]. We showed that with the increase in the concentration of palbociclib from 0.3µM to 10µM, the proportion of dead cells monotonically increased in comparison with the untreated control from 10.7% to 21.9%, respectively (Figure S1). Thus, the cytostatic effect of the inhibitor in the same concentration range led to an additional drop in the MTT signal by approximately 20-25% (Figure 1S).”

Point 2:   The authors should report in the Material and Methods sections how they calculate the EC50 and CC50 values in the pre-senescence model (in the presence of 1μM palbociclib).

Response 2:   We added the following text in section 4.6 of Materials and Methods – “Methods for measuring and calculating ЕС50 and СС50 values for DAAs and HDACi in pre-senescent cells were exactly the same as in the case of regularly proliferating cells”. It can be especially noted that in the control samples treated with palbociclib, the luciferase and MTT signals decreased by approximately 2 and 1.5 times, as shown in Figure 1 (D/E).

Point 3:  The authors reply regarding belinostat/vorinostat is still uncertain (obviously their anticancer effects is well known). In particular, it is not clear why they compared their antiviral activity in the pre-senescent cells with the cytotoxicity observed in regularly proliferating cells. I suggest revising the text (lines 509-515) with a focus on the reduced potency of their antiviral activity rather than their therapeutic potential.

Response 3:   This sentence was accordingly changed to - “The use of pre-senescent cells in the test system eliminated this problem due to the absence of proliferation, which ultimately made it possible for us to adequately measure the anti-HCV activity of these compounds and assess their antiviral potency under pre-senescence conditions which was unfortunately found to be significantly weakened”.

Point 4:    Title of Section 2.4 should be revised, as “Testing antiviral drugs under regular and pre-senescence conditions allows to distinguish between DAAs and HTAs” is not supported by the presented results in this manuscript.

Response 4:   The title of Section 2.4 was accordingly changed to “Comparative Testing DAAs and HDACi under regular and pre-senescence conditions”.

Point 5:    Surprisingly, the p-value (< 0.001) reported in Table2 indicates a significant difference in the TMP-269 antiviral activity in both cellular conditions, however in the text, it is reported that no difference was observed with this compound. Please revise it.

Response 5:    The original sentence was modified as follows: “The dose response of replicon inhibition by CI-994 and bufexamac or ortho-PhO-CHA, TMP-269 and tubastatin A was the same or even more sensitive in the reprogrammed test system compared to the regular one…” This version of the sentence contains the clarification proposed by the reviewer – “…was the same or even more sensitive …” – which simultaneously refers to three inhibitors – ortho-PhO-CHA, ТМР-269 and tubastatin А, as can be seen from Table 2.

Point 6:    Lines 33-35. The sentence introduced does not make sense. Please revise it.

Response 6:   A part of the sentence – “and address the challenges that lay ahead commonly associated with the viral resistance to treatment by DAAs” – was changed to – “and overcome future challenges associated with the emergence of resistance to DAAs” [1].

Point 7:    Line 72 (I simply meant to define the kind of controls in the sentence reported in line 72, “…had shown a sharp increase in the concentration of TGF-β1 in periportal hepatocytes and hepatic sinusoidal cells in comparison with control samples”) – Please define which kind of control (from healthy individuals?). Line 125 please refer to untreated control or solvent/vehicle control.

Response 7:    The sentence was changed according to the Reviewer’s suggestion – “…had shown a sharp increase in the concentration of TGF-β1 in periportal hepatocytes and hepatic sinusoidal cells in comparison with control samples from healthy individuals”. Instead of Line 125, the Reviewer probably meant Line 115; we made the corresponding clarification – “Thus, in comparison with the untreated control, both TGF-β1 and palbociclib addition resulted in an increase in the replicon level relative to the luciferase signal by 2-2.5 times.”

Point 8:    Please thoroughly revise English and technical language as some mistakes have been detected throughout the text. Revise also some sentences (e.g. lines 121-122).

Response 8:    Here is the new version of one sentence (Lines 123-125) – “Interestingly, according to the published data, palbociclib is able to induce proteasome activation mediated through the reduced proteasomal association of the ECM29 protein [37]”.

Reviewer 2 Report

In the current version the authors greatly improved their work. I profoundly appreciate their efforts in this respect. However I must recommend a few additional changes, particularly regarding their interpretation and study design relative to the claimed increase in lipid droplets biogenesis  experiments.

My comments are as follows

  • I think the authors misuse the term "correspondingly" in place of "respectively".  However I am not a mother tongue english speaker so I apologize if this is not the case. If so, I would suggest rewording.
  • Point 15:  All experiments. How are “untreated” cells treated? Does it involve ethanol or DMSO treatment? At which concentration. Please indicate such information or repeat experiments including appropriate vehicle treated cells.

    Response 15: Cell treatment protocol: the stock solution of the compound (usually 100 mM in DMSO) was serially diluted with alcohol to 100x concentration; 1% of the total volume was added to each well – 5 μl in case of a 48-well plate and 2 μl in case of a 96-well plate. Control samples in experiments with antiviral agents were treated with 1% v/v of ethanol. In the case of experiments with TGF-b1 and palbociclib, an appropriate volume of PBS was added to the control samples, since both substances were prepared and added to the wells as solutions in PBS.

    • I still do not understand how untreated cells were treated. Please provide in the text the exact final solvent concentration for each treatment (DMSO 0.05%, Ethanol 0.1%) and noter that the same final solvent concentration should be used for "untreated" cells
  • Point 17:  Figure 3. Please provide quantification of Lipid droplets and Lysosomes number and intensity as well as LDs and Ls co-localization (Pearson).

    Response 17: It is difficult to count the number of organelles because their images on cell sections overlap. In addition, there are dense clusters of the organelles. Therefore, in the new version, we measured the average area occupied by these organelles on the optical section of one cell (Figure 3A). Colocalization analysis was performed for two out of five parts of Figure 3В using Coloc2 and JACoP plugins. Image set CBS001RGM-CBS010RGM from the Colocalization Benchmark Source (www.colocalization-benchmark.com) was used to validate colocalization. Figure 3C presents the corresponding 2D cytofluorograms with values of Pearson’s coefficients. Corresponding changes were made to Materials and Methods.

    • please provide in the Materials and Methods the exact number of cells analyzed for colocalization and Pearson analysis
  • Figure 2A. Since the total of each group should be 100, maybe a pie representation is more indicated
  • Figure 3D. Please insert a legend for Figure 3D.
  • Heading 2.3 The authors demonstrate a decrease in lipophagy due to altered lipophagic flux. Therefore increased LD content (Figure 3D/E) and GFP-LC3 (Figure 4) do not necessarily reflect an "increased LD formation", but could simply due to an arrested autophagic flux. I suggest using "Dengue virus inhibition of autophagic flux and dependency of viral replication on proteasomal degradation of the autophagy receptor p62" by Metz et al., 2015 as a reference and including a bafylomycin treated cells as control. That would (should) show that simply blocking autophagic flux would increase GFP-LC3 punt and LD content. 

Author Response

Point 1:    I think the authors misuse the term "correspondingly" in place of "respectively". However, I am not a mother tongue English speaker so I apologize if this is not the case. If so, I would suggest rewording.

Response 1:    Both words, "correspondingly" and "respectively", mean the same.

Point 2:    I still do not understand how untreated cells were treated. Please provide in the text the exact final solvent concentration for each treatment (DMSO 0.05%, Ethanol 0.1%) and noter that the same final solvent concentration should be used for "untreated" cells

Response 2:    As far as we are informed in this matter, the expression “untreated cells” implies the absence of the tested substance and vehicle/solvent in the culture medium. As for DMSO or ethanol which are most often used as solvents for biologically active low-molecular-weight compounds, the presence of these organic solvents in control samples is assumed by default. According to our procedure, 100 mM stock solutions of tested compounds were prepared in DMSO, and after dilution with ethanol by the corresponding number of times, the DMSO content in 100x solutions decreased in a power-law manner (by approximately 10N/2 times, where N is the number of dilutions). Thus, the concentration of DMSO in the well was always different and always low. However, when determining the CC50 values, we often had to work with relatively high concentrations of compounds - 100mМ, 300mM and 1mM, when the content of DMSO in the well increased to 0.1% (plus 0.9% EtOH), 0.3% (plus 0.7% EtOH) and 1% (plus 0% EtOH), respectively. According to our observations, as well as according to data from literature, 1% DMSO does not have a significant effect on cell viability [Chan Yuan et al., 2014; Fatemeh Hajighasemi et al., 2017]; however, an increase in the DMSO content in the medium increases the permeability of cell membranes for low-molecular-weight compounds [Rebecca Notman et al., 2006]. As a consequence, there is an excessive accumulation of compounds in the cell and an underestimation of CC50 values. Unfortunately, the authors do not have information on how this effect can be neutralized or taken into account. Nevertheless, for greater transparency of the experimental results, and in order to meet the requirements of the Reviewer, we made an addition to section 4.2 of Materials and Methods: “The stock solutions of the tested compounds (usually 100 mM in DMSO) were serially diluted with ethanol to 100x concentrations; the corresponding 100x solution was added to each well of a culture plate in an amount of 1% of its volume; vehicle control samples were treated with the same amount of ethanol.”

Point 3:    Please provide in the Materials and Methods the exact number of cells analyzed for colocalization and Pearson analysis

Response 3:    To demonstrate the different effects of TGF- b1 and palbociclib on the colocalization of lipid droplets and lysosomes, we selected in both cases one of the most representative cells. Each of them had a high density and high intensity of green and red signals. Pearson's correlation coefficient was calculated for the selected fragments of these two cells, as described in the caption to Figure 3 (С): “Scatterplots of green and red pixel intensities (PI) of the cells shown in T3 (B) and P10 (B) and scatterplot in the region of interest (ROI) containing lysosomes in cells shown in T3 (B)”

Point 4:    Figure 2A. Since the total of each group should be 100, maybe a pie representation is more indicated.

Response 4:   The authors ask the reviewer to allow them to keep the original form of presentation of the results.

Point 5:    Figure 3D. Please insert a legend for Figure 3D.

Response 5:     The caption for the bars of the histogram was added to Figure 3D (red-LDs and green-Lysosomes).

Point 6:     Heading 2.3 The authors demonstrate a decrease in lipophagy due to altered lipophagic flux. Therefore, increased LD content (Figure 3D/E) and GFP-LC3 (Figure 4) do not necessarily reflect an "increased LD formation", but could simply due to an arrested autophagic flux. I suggest using "Dengue virus inhibition of autophagic flux and dependency of viral replication on proteasomal degradation of the autophagy receptor p62" by Metz et al., 2015 as a reference and including a bafilomycin treated cells as control. That would (should) show that simply blocking autophagic flux would increase GFP-LC3 punt and LD content.

Response 6:     The authors agree with the comment of the Reviewer that in relation to the action of palbociclib, the expressions "increased LD formation" and "accelerated biogenesis of LDs" appear excessive. In the new version of the manuscript, they were replaced everywhere with a neutral expression - "accumulation of LDs". We are grateful to the Reviewer for a relevant reference [Metz et al., 2015] and for the proposed control experiment. However, the authors believe that in the context of this work the analogy between palbociclib and bafilomycin will not be sufficiently justified, since the biological action of palbociclib relies on completely different mechanisms - primarily inhibition of CDK4/6 and chloroquine-like activity.
